# A deeper understanding of system interactions can explain contradictory field results on pesticide impact on honey bees

Dimitri Breda[1], Davide Frizzera [2], Giulia Giordano [3], Elisa Seffin[2], Virginia Zanni[2], Desiderato Annoscia [2], Christopher J. Topping [4], Franco Blanchini [1] ✉ & Francesco Nazzi [2] ✉

While there is widespread concern regarding the impact of pesticides on honey bees, well-replicated field experiments, to date, have failed to provide clear insights on pesticide effects. Here, we adopt a systems biology approach to gain insights into the web of interactions amongst the factors influencing honey bee health. We put the focus on the properties of the system that depend upon its architecture and not on the strength, often unknown, of each single interaction. Then we test in vivo, on caged honey bees, the predictions derived from this modelling analysis. We show that the impact of toxic compounds on honey bee health can be shaped by the concurrent stressors affecting bees. We demonstrate that the immune-suppressive capacity of the widespread pathogen of bees, deformed wing virus, can introduce a critical positive feed-back loop in the system causing bistability, i.e., two stable equilibria. Therefore, honey bees under similar initial conditions can experience different consequences when exposed to the same stressor, including prolonged survival or premature death. The latter can generate an increased vulnerability of the hive to dwindling and collapse. Our conclusions reconcile contrasting field-testing outcomes and have important implications for the application of field studies to complex systems.

Losses in honey bee colonies have been reported since the beginning of modern apiculture[1], but the scale of these events has increased dramatically[2]. These losses potentially affect pollination services and food sustainability[3] and are therefore a cause for concern. Losses are multifactorial with several interacting stress factors affecting honey bee health leading to potential cascade effects on colony stability[4]. Some of the factors which significantly contribute to colony losses are parasites and pathogens, agrochemicals, forage resource availability, and environmental conditions such as external temperature[5].

Pesticides, and in particular neonicotinoid insecticides, have attracted considerable attention for their potential negative effects on pollinators including honey bees[6]. These compounds have both lethal and sublethal effects on bees, affecting navigation, immunity, and reproduction[7-9]. However, even though the negative effects of neonicotinoid insecticides have been established in the laboratory[6], field testing has resulted in contradictory outcomes (Supplementary Table 1). No detectable negative effects were reported on honey bees maintained near Clothianidin-treated oilseed rape fields in some countries[10-12], whereas in a large-scale experiment spanning three

[1]Dipartimento di Scienze Matematiche, Informatiche e Fisiche, Università degli Studi di Udine, Udine, Italy. [2]Dipartimento di Scienze AgroAlimentari, Ambientali e Animali, Università degli Studi di Udine, Udine, Italy. [3]Dipartimento di Ingegneria Industriale, Università degli Studi di Trento, Trento, Italy. [4]Department of Ecoscience, Aarhus University, Aarhus, Denmark. ✉e-mail: franco.blanchini@uniud.it; francesco.nazzi@uniud.it

European countries both negative and positive effects were noted[13]. The lack of negative results observed in some cases has been attributed to the buffering capacity of honey bee colonies[12,14] but the reason why such buffering capacity could prevent apparent harm under certain conditions, but not others, remains unclear. The variability in the contexts where the studies were carried out, involving both the possible stress factors and the quantity-quality of available nutrition as well as the availability of other foraging resources in turn affecting the exposure to the pesticide applied to the focal crop, certainly plays a role. However, this plausible explanation lacks the necessary robustness in cases where the absence of evidence has often been regarded as evidence of absence. In fact, after several high-profile well-replicated experiments, the regulation across countries and regions with otherwise similar situations appears different in each. For example, in Europe, the neonicotinoids Clothianidin, Imidacloprid, and Thiamethoxam have been banned in open fields since 2018. While Canada banned the use of neonicotinoids on bee attractive crops in 2019 but still allows other uses including seed treatment and in the US a review on the same chemicals is still in progress. The situation is related to several factors, these include a different interpretation of the precautionary principle, economics, and politics. However, the consistency of scientific evidence provided to support such decisions may have played a role.

To gain a mechanistic understanding of the processes underlying the contrasting field results regarding pesticide harm to honey bees, we adopted a systems biology approach. Based on theoretical and computational parameter-free methodologies[15,16] we assessed the structural properties of the biological system under study (i.e., honey bee health as affected by various factors). These are properties that exclusively rely on the architecture of the system and are independent of the strength, which is often unknown, of each interaction (i.e., any relationship between two components of the system). Structural approaches can provide qualitative insight into complex webs of interactions, even in the absence of knowledge about parameter values, and unravel the synergistic net effect of multiple stressors on bee health. Through these methods we showed, first in theory and then in vivo, how the impact of toxic compounds on honey bee health and colony stability can be shaped by the concurrent stressors affecting bees, eventually leading to multiple outcomes depending on initial conditions.

## Results

### A conceptual model of honey bee health

The conceptual model of stressors and drivers potentially affecting honey bee health was built from available data (Fig. 1; Supplementary Table 2). This model describes the health of honey bees as influenced by multiple stress factors and effects. These include: (a) ectoparasites such as the mite *Varroa destructor*[17], (b) viral pathogens like the deformed wing virus (DWV)[18], (c) toxic compounds[19], among which neonicotinoid insecticides appear to play a pre-eminent role[6], and adverse environmental factors, in particular (d) sub-optimal thermal conditions[20]. Sugars from nectar (e) and pollen (f) are used by bees as a source of energy and proteins and promote honey bee health[21]. Both nectar and pollen can however be contaminated with toxic compounds (g, h)[22,23]. Honey bees invoke a number of mechanisms to combat stress factors; in particular, an immune response is normally activated to counter parasites (i)[24] and pathogens (j)[25], and a detoxification system (k) can reduce the concentration of toxic compounds[26]. Honey bees can increase sugar feeding to counteract low temperatures (l)[27]. However, this increased feeding may then expose bees to higher contamination with toxic compounds. Some of the factors themselves can influence honey bee homeostatic responses; DWV in particular can impair the immune response (m)[28], which can likewise be reduced by some toxic compounds (n)[7]. Mite-infested honey bees may consume less sugar (o)[29]. We also cannot discount that lower temperatures may have a potentially negative effect on parasites (p)[30].

Many more stressors, including more than twenty viruses, a plethora of toxic substances, several parasites, and a countless combination of environmental factors may influence bee survival[4]. However, as far as our analysis is concerned, the proposed representation of the system already captures all the relevant qualitative interactions, irrespective of the specific identity of the stressors involved and the quantitative details. For example, we included just one toxic compound, even though many pesticides can impact honey bees at the same time[31] and can interact with one another, as in the case of fungicides increasing the toxicity of insecticides[32–34]. Our model may thus be seen as an oversimplification of the system under study. This would be the case if our objective were to derive a descriptive model aiming

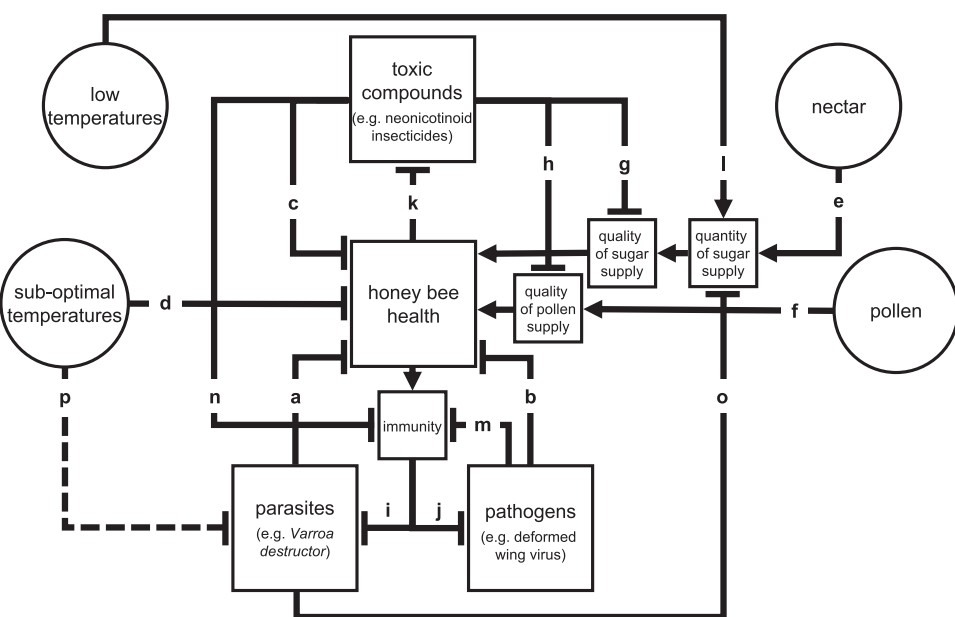

**Fig. 1 | The health of honey bees as influenced by multiple factors and their effects.** In the conceptual model of bee health bar-headed lines denote negative effects while arrow-headed lines indicate positive ones. See text and Supplementary Table 2 for explanation of lettered effects.

at quantifying bee health at any given time, in the presence of a defined level of certain stressors. However, for the structural analysis of the network of qualitative interactions we carried out, the case of one toxic compound exerting a negative effect, or that of more toxic compounds interacting with one another to exert an even bigger negative effect on honey bees, are equivalent because the sign of the effect is the same. Similarly, flowering resources can impact the way pesticide use affects bees either by deterring them from treated plants or by altering their pesticide tolerance[35–38]. However, such effects would correspond to a lower or higher impact of toxic compounds which have already been incorporated into the model. Thus, in both cases, the outcome of the analysis is not affected by this modeling choice.

According to our conceptual model, the dynamic interplay between honey bee health and the surrounding environment can be described by the following system of ordinary differential equations.

$$\tau_{HB}\dot{x}_{HB} = -\delta_{HB}x_{HB} + g_{TC}(x_{TC}) + g_{VA}(x_{VA}) + g_{VI}(x_{VI}) \\ + \bar{f}_{S,C}(u_S, u_C, x_{TC}, x_{VA}) + \bar{f}_P(u_P, x_{TC}) + \underline{f}_{HB}(u_T) \quad (1)$$

$$\tau_{TC}\dot{x}_{TC} = -\delta_{TC}x_{TC} + g_{HB}(x_{HB}) \quad (2)$$

$$\tau_{VA}\dot{x}_{VA} = -\delta_{VA}x_{VA} + h_{VA}(x_{HB}, x_{TC}, \varepsilon x_{VI}) + \underline{f}_{VA}(u_T) \quad (3)$$

$$\tau_{VI}\dot{x}_{VI} = -\delta_{VI}x_{VI} + h_{VI}(x_{HB}, x_{TC}, \varepsilon x_{VI}) \quad (4)$$

These equations mathematically represent the interactions among the key components (variables) in our conceptual model: honey bee health ($x_{HB}$), the stress due toxic chemicals ($x_{TC}$), the stress due to parasites ($x_{VA}$), and the stress due to pathogens ($x_{VI}$). The system includes the effects of the external inputs: sugar $u_S$, pollen $u_P$, absolute deviation from desired temperature $u_T$ and sub-optimal temperature $u_C$. The coefficients $\tau$ denote the time constants, $\delta$ denote the "self-control" of each key-player. All inputs (and possible parameters, e.g., $\varepsilon$) are non-negative. All variables and inputs exert their influence on the variation of the other variables (denoted by a dot on the variable's name) by means of different functions (i.e., $g(x)$, $\bar{f}(x)$, $\underline{f}(x)$, $h(x)$). Functions can be decreasing, in case of negative effects (e.g., function $g_{TC}(x_{TC})$ in Eq. (1), representing arrow c in Fig. 1, indicates that the more toxic compounds $x_{TC}$, the lower honey bee health $x_{HB}$). Functions can also be increasing/decreasing according to the variable or input considered (e.g., function $\bar{f}_P(u_P, x_{TC})$ in Eq. (1), representing arrow f in Fig. 1; the function is increasing with respect to $u_P$, because the more pollen the higher honey bee health, but is decreasing with respect to $x_{TC}$, because toxic compounds can contaminate the pollen and thus cause a negative effect on honey bee health (see arrow h in Fig. 1)). A detailed description of the various functions, together with a summary of the biological effects they account for and a reference to the conceptual model in Fig. 1, is reported in Supplementary Table 3.

Equation (1) shows that honey bee health ($x_{HB}$) is self-regulated by internal physiological mechanisms described by $\delta_{HB}$. Also, honey bee health can be negatively influenced by toxic compounds ($x_{TC}$), parasites ($x_{VA}$), and pathogens ($x_{VI}$), according to various mechanisms described by different monotonically decreasing functions (i.e., $g_{TC}$, $g_{VA}$, $g_{VI}$), denoted with the common symbol $g$ because each factor exerts a negative effect on honey bee health. Similarly, honey bee health is affected by other factors (e.g., nutrition, represented by the external inputs $u_S$ and $u_P$; sub-optimal temperatures $u_T$ and low temperatures $u_C$ whose influence can be modified by other stress factors (e.g., toxic compounds that can contaminate foodstuff). These interactions are represented by functions that are increasing in the case of favorable influences and decreasing in the case of adverse effects.

## Structural analysis of the bee health model

The structure of the system under study (i.e., honey bee health as affected by various factors) was analyzed using the concept of community matrix[39]. The community matrix, whose elements represent the effects of each factor onto every other and itself at equilibrium, formally encapsulates the interactions among the components of an ecological system and corresponds to the Jacobian matrix of the system of growth equations, together with their respective signs. Since the signs of the partial derivatives for the various functions are as described above, if we assume that the negative term $\delta_{VI}x_{VI}$ is dominant with respect to the positive effect from $h_{VI}$, then the Jacobian matrix of the system has the following parameter-independent sign pattern, where the term in position ($i,j$) represents the parameter-independent sign (positive, negative, or zero) of the direct effect that key player $j$ has on key player $i$.

$$sign(J) = \begin{bmatrix} - & - & - & - \\ - & - & 0 & 0 \\ - & + & - & + \\ - & + & 0 & - \end{bmatrix}$$

If the model is reformulated by using as a first state variable the opposite of $x_{HB}$ (viz. an indicator of bee unhealthiness), the community matrix becomes Metzler (i.e., all off-diagonal entries are non-negative); hence, the system is monotone[40]. Monotonicity consists in the remarkable feature of preserving the ordering of solutions with respect to initial data. When this is the case, despite the possible intricacies, some important features of the system dynamics can be inferred based on purely qualitative or relatively basic quantitative knowledge of the system characteristics[41,42] as we will show below.

We then described the effect of an external input applied to the system variables on the steady-state variation of each of the others. If a persistent input is applied to the system, the steady-state variation of a variable may have the same sign as the applied input, or the opposite sign, or may be zero in the case of perfect adaptation. Structural influence means that the sign of the variation does not depend on the value of the system parameters. In this case, the steady-state interactions can be represented by the following structural influence matrix, where the term in position ($i,j$) represents the parameter-independent sign (positive, negative, or zero) of the variation of the steady state of key player $i$ ensuing from the application of a constant input affecting key player $j$; this can be seen as the net effect of $j$ on $i$, including both direct and indirect effects. HB, TC, VA, VI are honey bee health, toxic compounds, parasites, viruses, respectively.

| Influence of | HB | TC | VA | VI |
|---|---|---|---|---|
| on HB | + | − | − | − |
| on TC | − | + | + | + |
| on VA | − | + | + | + |
| on VI | − | + | + | + |

Unlike the sign matrix above, which includes only the direct effect of each component on the others, this matrix reports net effects, including both direct and indirect effects of a stressor on the others[16]. The structural influence matrix thus shows that any new stressor applied to the system has a net negative effect on bee health. Thus, a toxic compound, such as for example a neonicotinoid insecticide, can only have a negative effect on honey bee health when applied to individual bees, regardless of the presence of parasites and pathogens. Hence, the lack of a detectable effect reported in some cases could be regarded as a lack of the hypothesized detrimental effect. However, a detailed study of the system equilibria reveals that this conclusion is a consequence of not considering the complexity of the study system (i.e., honey bee health as affected by various factors).

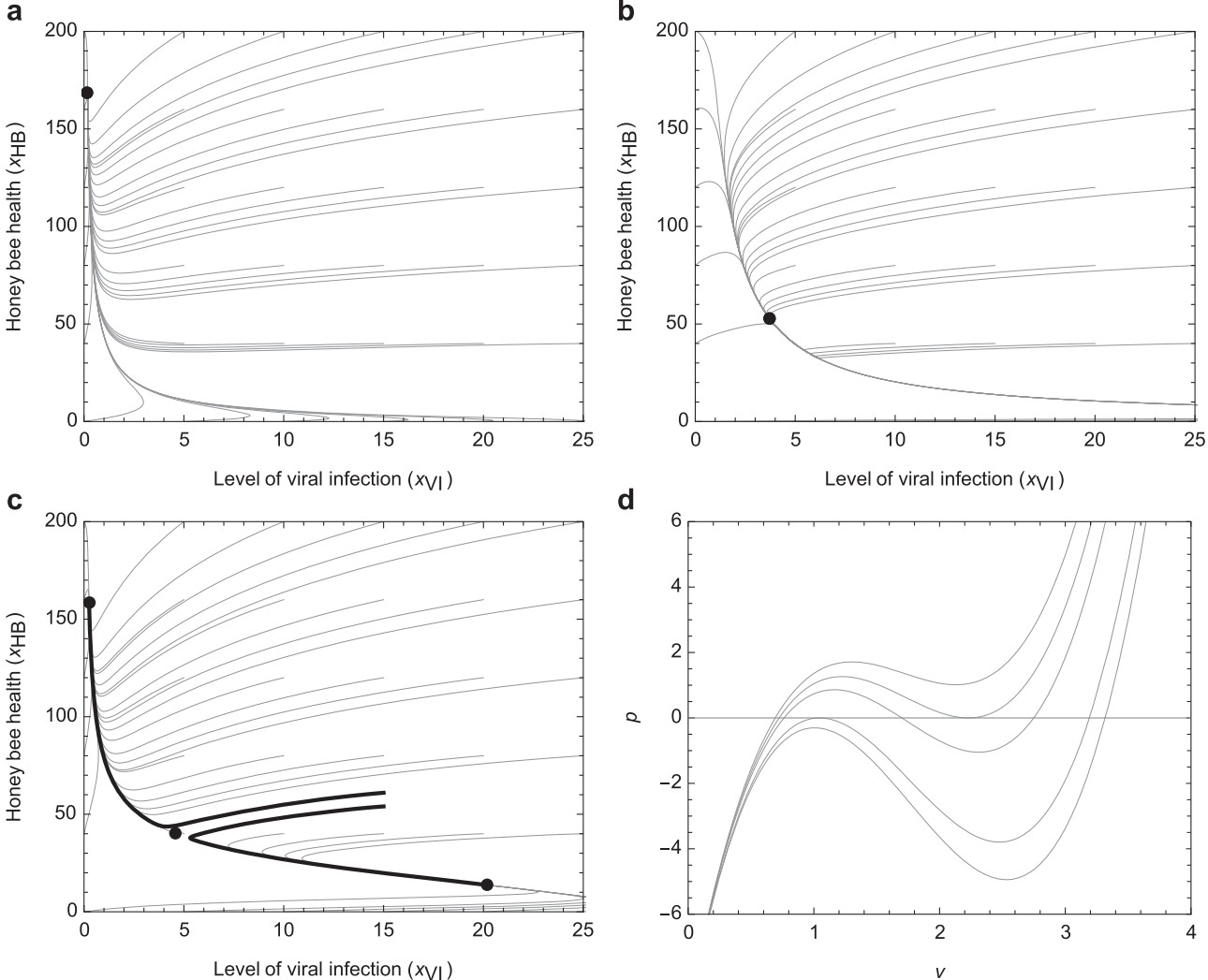

**Fig. 2 | The equilibria and some orbits of the full system in the projected phase plane of honey bee health ($x_{HB}$) and level of viral infection ($x_{VI}$).** Equilibria represent the values of the state variables where they do not change and are indicated with dots, while the orbits are the values that the state variables can assume while approaching the equilibria and are represented with lines. **a** Orbits and the unique equilibrium without immune-suppression, in presence of a low level of parasites. **b** Orbits and the unique equilibrium without immune suppression, in case of a high level of parasites. **c** Orbits and the three equilibria with immune-suppression; two orbits exiting from close initial conditions are marked with thick lines. **d** Equilibria of the subsystem of bee and virus for increasing immune-suppression. $p$ is a function of the level of viral infection $v$ that vanishes at equilibria; top curve: at low immune-suppression there is one equilibrium at high bee health; bottom curve: at high immune-suppression there is one equilibrium at low bee health; intermediate values of immune-suppression can cause three equilibria.

## System equilibria

Although an analytical solution of the differential equations representing our biological system, and thus the calculation of each variable at each time, is not possible, the study of the equilibria of the system can explain its behavior under different conditions.

Equilibria are the simplest solutions of the dynamical system representing honey bee health as affected by stressors and drivers and represent the value of the state variables (e.g., $x_{HB}$, representing honey bee health) where they do not change, or, in other words, the possible destiny of a variable provided it is allowed to (and can) settle to a constant value. Therefore, the study of system equilibria can discriminate whether honey bee health, represented by Eq. (1), can settle to a high, satisfactory level, or is bound to deteriorate to a lower, dangerous level, when insects are exposed to a certain set of stressors. The equilibria and the orbits (i.e., the values that the state variables can assume while approaching equilibria) are represented in graphs with black dots and lines, respectively (Fig. 2a–c).

To provide a visual description of our results, we specified the form of each function and assumed a set of values for the model parameters (Supplementary Methods; Supplementary Table 4); then we plotted the orbits and the equilibria on the projected phase planes. In this way we could graphically describe the trajectory of each variable with respect to others; in particular, we could see how honey bee health reacts to increasing pressure of viruses, parasites or toxic compounds and the end point of this process. Please note that our arbitrary selection of parameters (which are highly uncertain) does not influence the general qualitative conclusions of this study.

To investigate stability in the presence of different stressors, we considered two alternative cases: (1) a pathogen that cannot influence the immune response of honey bees, and (2) a viral pathogen that can affect honey bee immune system, as in the case of DWV[28].

In the first case, after appropriate mathematical treatment (Supplementary Methods), we found that the system admits a unique positive equilibrium, which is globally asymptotically stable in the positive orthant, whereby the position of the equilibrium on the honey bee axis depends on the intensity of the stressors or their combination (Supplementary Results; Fig. 2a, b). In particular, in presence of a pathogen that cannot impair immunity, honey bee health is high when

the level of parasites (or any other stressor) is low (Fig. 2a), and vice versa (Fig. 2b). In other words, it appears that, in presence of a stable input of the stressors included in our model, honey bee health reaches a well-defined level which depends on the level of the stressors. If either the level of parasite or pathogen pressure or pesticide contamination or both is too high, this equilibrium can be unbearable for the individual bee, resulting in death. In any case, the result can be predicted with a good degree of confidence based on the initial conditions; in fact, global stability makes the result independent of the initial conditions, as highlighted by the orbits in Fig. 2a, b that are converging to the same equilibrium point (represented by the dots in the figures) from different initial conditions (represented by any point on the lines in the figures).

We then considered the presence of a pathogen with the capacity to affect the immune response of honey bees. In this case, a convenient mathematical treatment relying also on bifurcation theory[43] (Supplementary Methods) reveals a completely different scenario: the system can now admit three equilibria, one of which is unstable, and hence bistability arises (Fig. 2c). A dynamical system is bistable when it has got two stable equilibria. This is a common feature of many biological systems and allows to interpret several phenomena from the level of molecules to ecosystems (see for example, refs. 44–47). With a convenient metaphor, a monostable system (i.e., a system with a single stable equilibrium, like the one described above) can be assimilated to a landscape with a single valley such that a ball will inevitably end at the bottom of that valley. Instead, a bistable system can be represented with two valleys separated by a hill, such that a ball sitting on the top of the hill (i.e., in the unstable equilibrium) can either fall into one or another valley, depending on any small initial perturbation.

Bistability is related to the presence of positive feed-back loops in the system that can amplify small differences in the initial conditions[48]. In this case, the addition of a pathogen that is capable of interfering with the immune response corresponds to the introduction of a critical positive feed-back loop into the system (formed by arrows "m" and "j" between "immunity" and "deformed wing virus" in Fig. 1). Indeed, the higher the viral load, the stronger the suppression of the immune system, and the lower the efficiency of the latter to contain the virus, which can then actively replicate leading to higher viral loads. In mathematical terms, this can be seen from the equations of the system: functions $h$, which convey the effect of the virus, are increasing with respect to $x_{VI}$ (the state variable associated with the virus). When the parameter $\varepsilon$, associated with the immune-suppressing potential of the virus, is large enough, the presence of function $h_{VI}$ in the equation describing the time evolution of $x_{VI}$ yields the ability of the virus to increase its effect. Thus, the presence of an immune-suppressing virus creating a positive feed-back loop is necessary for the system to exhibit the described bistability property.

In practice, under reasonable and biologically meaningful conditions, if the immune suppression capacity is absent or low, a unique stable equilibrium exists in the range of high bee health (Fig. 2d). For higher immune-suppression (i.e., larger values of the crucial parameter $\varepsilon$ in Eqs. (3) and (4)) a fold bifurcation[43] creates two additional equilibria (Fig. 2d). Of the resulting three equilibria, two are stable and are located in the high and low bee health regions, respectively. Increasing $\varepsilon$ further moves the intermediate unstable equilibrium towards the high bee health stable one, until they collapse and disappear through a second fold bifurcation, leaving just one stable equilibrium in the low bee health region, when the immune suppression capacity is too large (Fig. 2d).

In conclusion, the introduction of a pathogen capable of interfering with the honey bee's immune system generates an unstable intermediate 'watershed' equilibrium, which explains why, in the presence of slightly different initial conditions, vastly different outcomes can be possible (see thick curves in Fig. 2c). Under more descriptive terms, if a stressor is above a certain level, there is only one equilibrium at low bee health, meaning for example that if a toxic compound is present at a harmful concentration, bee survival will be significantly lower, and a negative effect will be noted; instead, if the same stressor is below that dangerous level, one equilibrium at high bee health is certainly possible; meaning that, if the toxic compound is present at a low concentration, bee survival may not be significantly different from normal and a negative effect may not be noted. Interestingly, our analysis revealed that, in the presence of an immune-suppressing virus, bistability can occur so that, for the same intermediate level of one stressor, one can have either low bee health or high bee health depending on the similar, but not identical, initial conditions and therefore the results may become unpredictable. In other words, in the presence of an intermediate amount of insecticide, a virus-infected bee can either die prematurely or survive much longer, depending on its initial, intrinsic individual situation.

## Validation of the bee health model

To experimentally test the predictions of our mathematical analysis showing bistability, we used data from several survival experiments, carried out using the same standardized method, over 6 years.

In this case, to test our theoretical predictions we used the longevity of caged bees as an estimator of their health condition, assuming that high honey bee health implies normal survival and low bee health is related to shorter longevity. Furthermore, to determine the effects of an immune-suppressing pathogen on honey bee health we used the seasonality of a common virus, DWV, which, in the area where the study was carried out, is rare in Spring and widespread at the end of the season when high viral loads are normally reached in infected bees[28]. For this reason, bees sampled early in the season can be considered virtually virus free whereas bees sampled late in the season can be considered as virus infected.

We hypothesized that, in the presence of an immune-suppressing pathogen, besides the expected reduction in median survival, the predicted bistability should result in bees at high bee health dying later in life and bees at low bee health dying earlier in life, with a consequent increase in the variability of longevity data.

We first tested the effect that the addition of an immune-suppressing virus has on the survival of caged honey bees. To this aim we compared the survival of bees maintained under the same conditions and sampled either early in the season and late in the season; subsequent qRT-PCR analyses confirmed that virus infection was rare in the first and common in the latter (Table 1; Supplementary Figure 1). Virus-free honey bees from early year populations had a characteristic survival curve with limited mortality during the first three weeks of life, followed by another two weeks of increased mortality with a distribution of lifespans centered around 23 days of age; in fact, 50% of those bees died between 21 and 24 days of age (Fig. 3a; Table 1). Instead, virus-infected honey bees from late-year populations had a shorter median survival and moreover a much broader distribution of lifespans, with a significant number of bees dying at a young age and others surviving much longer (Fig. 3a; Table 1). As a result, the interquartile range of longevities, here used as a measure of the dispersion of data, was 6 in early year bees and 10 in late year populations (Table 1), indicating a higher variability of longevity data in the presence of an immune-suppressing virus.

In a second experiment, virus-free honey bees were artificially fed virus particles or not and the tests were repeated, confirming the results reported above (Fig. 3b; Table 1). In particular, we found that control bees had a median longevity of 18 days and an interquartile range of 5, whereas virus-treated bees had a shorter median longevity (i.e., 10) as a result of a large number of bees dying in the first days, as underlined by a much larger dispersion of longevity data (interquartile range = 12). This further supports the view that the presence of an immune-suppressing virus can create vastly different outcomes depending on the slightly different initial conditions of single bees exposed to otherwise identical situations.

**Table 1 | DWV infection and survival of the honey bees used in the lab experiments**

| Treatment | Early | | | | | | | | | | Late | | | | | | | | | |
|---|---|---|---|---|---|---|---|---|---|---|---|---|---|---|---|---|---|---|---|---|
| | Control | | | | | Treated | | | | | Control | | | | | Treated | | | | |
| | DWV prev. | nv | Median survival | IQR | ns | DWV prev. | nv | Median survival | IQR | ns | DWV prev. | nv | Median survival | IQR | ns | DWV prev. | nv | Median survival | IQR | ns |
| None | 0.09 | 11 | 23.0 | 6.0 | 107 | | | | | | 0.70 | 63 | 21.0 | 10.0 | 542 | | | | | |
| Virus | 0.38 | 8 | 18.0 | 5.0 | 37 | 0.75 | 8 | 10.0 | 12.0 | 38 | | | | | | | | | | |
| Nicotine | 0.00 | 3 | 28.0 | 5.0 | 37 | 0.00 | 3 | 25.0 | 7.0 | 37 | 0.83 | 12 | 14.0 | 11.5 | 51 | 0.83 | 12 | 14.0 | 8.0 | 55 |
| Low temperature | 0.33 | 3 | 23.0 | 3.0 | 61 | 0.33 | 3 | 19.5 | 4.0 | 54 | 0.87 | 31 | 18.0 | 12.0 | 201 | 0.88 | 34 | 17.0 | 16.0 | 217 |

DWV prevalence (proportion of infected bees in a sample of nv bees), median survival (days), interquartile range of the distribution of longevities (IQR), and sample size (ns), are reported.

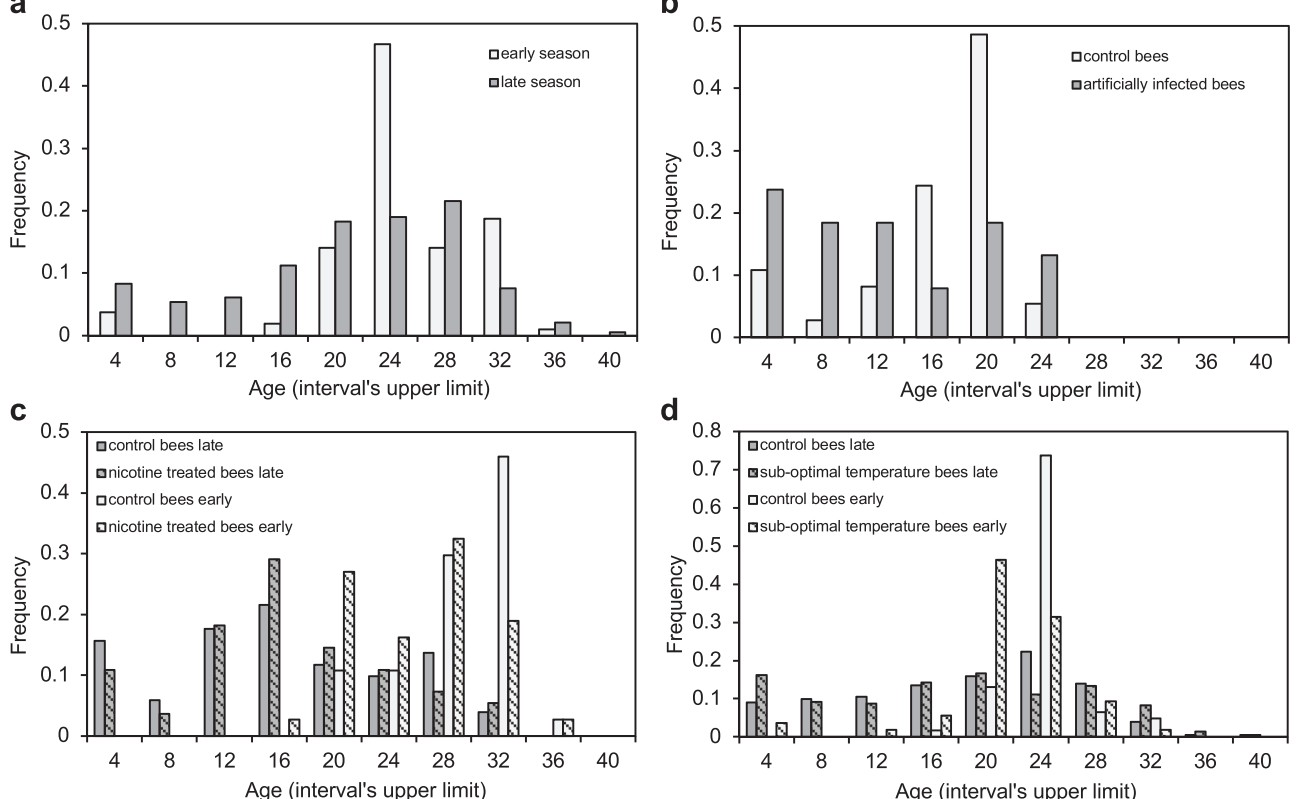

**Fig. 3 | Distribution of individual lifespans of honey bees under different conditions. a** Early in the season when the prevalence of an immune-suppressing virus is low (white bars) and later when all bees are virus infected (gray bars). **b** Treated or not (gray and white bars, respectively) with a virus administered to mature larvae through the diet. **c** When exposed to a toxic compound, when the prevalence of an immune-suppressing virus is low (white bars with diagonal pattern) or when the virus is widespread (gray bars with diagonal pattern); the corresponding distribution of honey bees sampled early or late in the season and not exposed to the toxic compound as a control (white and gray bars, respectively). **d** As (**c**) but exposed to a sub-optimal temperature in place of a toxin. Source data are provided as a Source data file.

In summary, by carrying out two different comparisons of uninfected versus virus-infected bees (one diachronic, with naturally virus-infected bees sampled at two different times, and one synchronic, by treating or not with the virus some uninfected bees at the same time), we noted that uninfected bees show mortality concentrated after three weeks of life, as expected given the shape of the survival curve of control caged bees previously observed under the same conditions[49]. In contrast, the mortality of virus-infected bees is not concentrated late in life but can also occur at a young age, resulting in a marked variability of longevities. Thus, as predicted by our model analysis, the probability of dying either soon or late does not only depend on the treatment but rather on the slightly different intrinsic conditions of bees. These were not under our control but dictated the bee's final destiny.

To investigate how the presence of an immune-suppressing virus could alter the response of honey bees to different stressors, we carried out two more experiments, whereby we studied the survival of honey bees exposed to 50 ppm of nicotine, here used as an example of a toxic compound, or to the sub-optimal temperature of 32 °C, as compared to the normal in-hive temperature of 34.5 °C[20].

When the virus was not present, both stressors caused a decreased lifespan, showing a distribution of lifespans shifted towards shorter ages (Fig. 3c, d; Table 1). However, in presence of a virus, both in the case of a toxic compound and a low temperature, a much broader survival distribution was generated, consistent with the bistability hypothesis (Fig. 3c, d; Table 1). Accordingly, the interquartile range of longevities increased from values from 3 to 7 in early year populations to values from 8 to 16 in late year populations (Table 1),

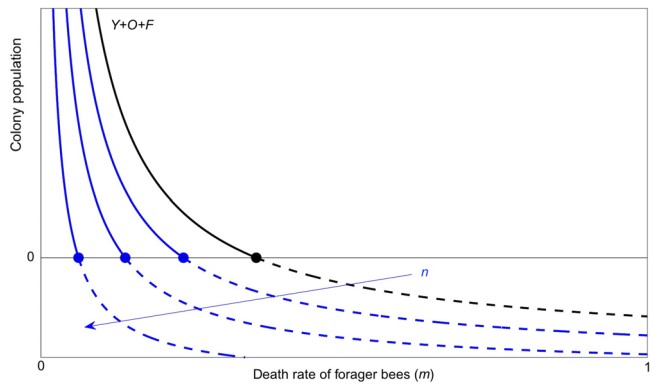

**Fig. 4 | Dependence of the colony population at equilibrium on the death rate m of forager bees for varying death rate n of juveniles hive bees.** For a forager death rate $m$ exceeding a critical value (black dot) the only stable equilibrium is zero, corresponding to colony failure. The premature death of hive bees (denoted by increasing values of $n$, represented by the blue curves) moves that critical value left, meaning that colony failure can occur for lower foragers' death rates. Black line: $n = 0$, blue: $n \in (0, 1)$, dots: $m(n)$. The parameter values are $L = 2000$, $w = 27{,}000$, $\alpha = 0.25$, and $\sigma = 0.75$ as in a previously published report[50].

highlighting a higher variability of longevity data, both in case of a toxic compound and a low temperature.

Overall, these results show that the presence of a pathogen capable of interfering with immune control creates a situation whereby the survival of honey bees is not solely determined by external stressors. Rather, it is greatly influenced by some minimal variations in the starting conditions, leading either to an imbalanced condition and premature death (lower thick orbit in Fig. 2c), or coping with the stress much longer (upper thick orbit in Fig. 2c).

**From individual health to colony stability**

It is important to note that any stress impacting the health of individual honey bees, thus significantly reducing their survival, could be propagated at colony level, eventually leading to colony collapse. However, whilst a mild negative effect could be buffered by the bee colony, a deviation from a favorable initial condition could result in rapid deterioration.

It has previously been shown that the lowered survival of forager bees can disrupt the colony equilibrium, resulting in colony collapse[50]. In particular, it was shown that mortality of forager bees exceeding a certain threshold (i.e., $m = 0.355$ in Fig. 4) could lead to colony failure despite some compensation mechanisms (e.g., a premature transition to foraging by nurse bees to replace dead foragers). To understand how the effects observed here in individual bees can influence colony stability, we used the same model after appropriate modifications. We found that the premature death of bees, at a younger age, as we report above, can be more detrimental than the already demonstrated reduced lifespan of foragers, moving the critical value of mortality to the left (Fig. 4). This mortality limits both the development of brood and the replacement of dying forager bees, adding to the effect described by other studies, and making collapse even more probable.

## Discussion

It is widely acknowledged that agricultural systems function as complex systems and agrochemicals are an important component within these systems. In particular, widely used neonicotinoid insecticides are regarded as significant threats to honey bees and the pollination service they provide, benefitting crop production and biodiversity[51]. This concern is based on a large and consistent body of evidence that was largely built under laboratory conditions[6]. Studies carried out under field conditions have not provided similarly convincing data[10-13]

(Supplementary Table 1), generating uncertainty about the real risk posed by some substances under more realistic settings. This, in turn, contributed to different regulatory approaches towards the same products under different conditions or countries[52].

Indeed, descriptive models could help draw more or less accurate predictions regardless of the inevitable variability of contexts[53] and thus support risk assessment and the consequent decisions[54]. Unfortunately, the lack of exact quantitative knowledge of the many parameters influencing bee health at individual and colony levels still pose a serious challenge to this approach. On the other hand, theoretical and computational tools are now available to assess the parameter-independent, structural properties of biological systems[15,16]. In fact, our systems biology approach allowed us to uncover some structural properties of the system under study (i.e., honey bee health as affected by various factors), reaching important conclusions that are based on unequivocal mathematical arguments.

We demonstrated in theory, and also confirmed in practice, that the already reported capacity of a widespread virus to impair the immune defenses of honey bees[28] can generate bistability. This implies that honey bees under similar initial conditions can have markedly different destinies when exposed to the same stressor. Our study of the possible consequences of this phenomenon at the colony level indicates that it increases the vulnerability of the colony to dwindling and collapse.

It is important to underline that only the immune-suppressing pathogen can cause the bistability and the described dynamics, because of its capacity to attack the bee defense system, thus exacerbating the pathogen's effect[28]. To our knowledge, no other stress factor can impair the system keeping that stressor under control and thus be implicated in similar dynamics. In some cases, an effect of pesticides on the detoxification system of honey bees has been reported[55]. This is normally expressed as an upregulation of some genes after exposure to pesticides[56-58], likely indicating the activation of a pathway in response to intoxication. This does not necessarily suggest the capacity of that pesticide to impair detoxification but can be regarded as evidence of a well-functioning homeostatic system that reacts to intoxication through a physiological mechanism aimed at reducing the concentration of the toxic chemical. On the other hand, several studies showed that fungicides can increase the toxicity of insecticides[32-34] suggesting impaired detoxification that could be tested with further mechanistic studies. Based on our analysis we can hypothesize that a pesticide exhibiting an anti-detoxification activity could cause system behavior like that reported here for the pathogenic virus DWV. At present this possibility is purely speculative, but it may have important implications for honey bee survival and should therefore be considered with great attention.

Our data allows a retrospective evaluation of published studies that may explain the contrasting results reported. Based on our conclusions we hypothesize that, in the presence of a low prevalence of the immune-suppressing virus, the negative effect of pesticides at field-realistic concentrations can be buffered by the colony's homeostatic response as previously proposed[12,14], provided that other stressor effects are limited. In contrast, when the immune-suppressive virus or the vector mite is present, negative effects are more likely to be observed because of the bistability we demonstrated. This in turn may cause some bees to experience premature mortality which cannot be effectively buffered by the homeostatic response mechanisms of the colony. This concurs with the observation that, in studies that showed no adverse neonicotinoid effects[10-12], DWV prevalence and/or mite infestations were low. Whereas, in the study reporting a country-specific effect of neonicotinoids[13], mite prevalence was low where positive effects were found (i.e., Germany) and high where effects were clearly negative (i.e., United Kingdom). Based on our results we suggest that the relative scarcity of the immune-suppressing virus can account for dynamics characterized by a single stable equilibrium at

satisfactory honey bee health. Under these conditions, it is likely that the buffering capacity of the bee colony can prevent collapse, despite a chemical reducing the bees' lifespan. If the immune-suppressing virus reaches a sufficient prevalence, the ensuing bistability accounts for results that can be either normal, when initial conditions are favorable, or dramatic in all other cases. This does not hold for neonicotinoid insecticides only, but also for parasites and pathogens, not least because *V. destructor* has allowed DWV to spread worldwide[59]. Of course, other factors, such as variable pesticide exposure under different conditions may be implicated in the variability of results about neonicotinoid effects under field-realistic conditions. However, in our opinion, such possible alternative causes would hardly result in a situation where the same chemical can cause either positive or negative effects in a comparative study carried out in a standardized manner[13].

In science we often rely on empirical data to base our predictions of future effects. This approach works well for limited and easily controlled systems, but it is not adequate for complex systems such as agroecosystems. Here, with honey bees, we show how even a small part of such a system can generate complex yet predictable emergent properties that can explain hitherto hard-to-reconcile observations.

Overall, this study demonstrates that considering relationships between components, rather than focusing on the individual, context-dependent, expression of a system state, leads to a much deeper understanding and is a better basis for real-world decisions. In fact, the bee system described here is a good example of the kind of feedbacks found in ecology and biology and is not unique. In cases like this, empirical observations of a single system state in space and time are important but have poor predictive power compared to the system analysis presented here.

Here, we demonstrate that although the complexity of the system representing honey bee health as affected by multiple factors can appear intractable, it may be better to deal with that complexity rather than to factor it away. This thinking suggests more critical evaluation of empirical studies and should help to clarify the debate on pesticides and honey bees. Today's regulatory risk assessment for pesticides relies on a single substance, single-use approach[54], but a new multi-stressor approach is proposed[60]. In parallel, discussions about the protection goals for bees in European environmental risk-assessment seem almost entirely based on empirical observation of variability, and not on mechanistic understanding[61]. Our results could inform regulatory efforts by contributing to re-design honey bee risk assessment and achieve a more homogenous regulatory response to scientific evidence.

# Methods

## The bee health model

The conceptual model of the interactions of various stressors with honey bee health is described by the following system of ordinary differential equations (ODEs)

$$\tau_{HB}\dot{x}_{HB} = -\delta_{HB}x_{HB} + g_{TC}(x_{TC}) + g_{VA}(x_{VA}) + g_{VI}(x_{VI}) \\ + \bar{f}_{S,C}(u_S, u_C, x_{TC}, x_{VA}) + \bar{f}_P(u_P, x_{TC}) + \underline{f}_{HB}(u_T) \tag{1}$$

$$\tau_{TC}\dot{x}_{TC} = -\delta_{TC}x_{TC} + g_{HB}(x_{HB}) \tag{2}$$

$$\tau_{VA}\dot{x}_{VA} = -\delta_{VA}x_{VA} + h_{VA}(x_{HB}, x_{TC}, \varepsilon x_{VI}) + \underline{f}_{VA}(u_T) \tag{3}$$

$$\tau_{VI}\dot{x}_{VI} = -\delta_{VI}x_{VI} + h_{VI}(x_{HB}, x_{TC}, \varepsilon x_{VI}) \tag{4}$$

for the state variables $x_{HB}$ representing honey bee health, $x_{TC}$ the stress due to toxic compounds (e.g., neonicotinoid insecticides), $x_{VA}$ the stress due to parasites (e.g., *V. destructor*) and $x_{VI}$ the stress due to

pathogens (e.g., DWV). The system includes the effects of external inputs as sugar $u_S$, pollen $u_P$, absolute deviation from desired temperature $u_T$ and sub-optimal temperature $u_C$. All the inputs and possible parameters are non-negative; the coefficients $\tau$ denote the time constants; the coefficients $\delta$ denote the self-regulation parameters; $\varepsilon$ in the last two equations allows to account for pathogens that can ($\varepsilon > 0$) or cannot ($\varepsilon = 0$) impair the immune system (through link m in Fig. 1). We assume that the functions $g$ are smooth, bounded, positive, convex and decreasing to 0; the functions $\bar{f}$ are smooth, bounded, non-negative, concave and increasing with respect to (w.r.t.) $u$ arguments (vanishing only when the first $u$ argument vanishes) while convex and decreasing to 0 w.r.t. $x$ arguments; the functions $\underline{f}$ are smooth, bounded, non-positive and decreasing (vanishing only when $u = 0$); the functions $h$ are smooth, bounded, positive, convex and decreasing to 0 w.r.t. the first argument while concave and increasing w.r.t. all the other arguments. For a detailed description of the various functions, together with a summary of the biological effects they account for and a reference to the conceptual model in Fig. 1, see Supplementary Table 3.

## Structural analysis of the bee health model

We describe here the structural considerations and computations that yield the structural influence matrix for the honey bee health system.

The structural influence matrix $M$ is defined as follows. $M$ is a symbolic matrix with entries $M_{ij}$ chosen among: $+, -, 0, ?$, according to the criteria described below. Consider an equilibrium point $\bar{x}$ and a constant perturbation $u$ applied on the $j$-th system variable (small enough not to compromise the stability of the equilibrium). The equilibrium value will be modified as $\bar{x} + \delta\bar{x}$. Consider the sign of the perturbation of the $i$-th variable, $\delta\bar{x}_i$. Then $M_{ij} = +$ if $\delta\bar{x}_i$ always has the same sign as $u$; $M_{ij} = -$ if $\delta\bar{x}_i$ always has the opposite sign as $u$; $M_{ij} = 0$ if always $\delta\bar{x}_i = 0$; regardless of the system parameters. Conversely, if the sign does depend on the system parameters, we set $M_{ij} = ?$.

In this section we prove that the influence matrix of the honey bee health system is structurally determined, i.e., there are no "?" entries in $M$.

We start with the following proposition.

*Proposition 1 Assume that a matrix J is Hurwitz stable (i.e., all its eigenvalues have negative real part) and has the sign pattern*

$$sign(J) = \begin{bmatrix} - & - & - & - \\ - & - & 0 & 0 \\ - & + & - & + \\ - & + & 0 & - \end{bmatrix}$$

*Then, the sign pattern of adj(−J), the adjoint of −J, is*

$$sign(adj(-J)) = \begin{bmatrix} + & - & - & - \\ - & + & + & + \\ - & + & + & + \\ - & + & + & + \end{bmatrix}$$

**Proof** To prove the statement, we just change the sign of the first variable, hence we change sign to the first row and column of matrix $J$. The resulting matrix $M$ is such that

$$sign(M) = \begin{bmatrix} - & + & + & + \\ + & - & 0 & 0 \\ + & + & - & + \\ + & + & 0 & - \end{bmatrix}$$

We observe that $M$ is a Metzler matrix, namely, all its off-diagonal entries are non-negative. Moreover, the matrix is Hurwitz stable. Then, we can proceed as in the proof of *Proposition 4* in a previous report[16]. Given a Metzler matrix that is Hurwitz stable, its inverse has non-

positive entries; hence, the inverse of $-M$ has non-negative entries: $(-M)^{-1} \geq 0$ elementwise. Moreover, we observe that $M$ is an irreducible matrix, i.e., there is no variable permutation that brings the matrix in a block (either upper or lower) triangular form. This implies that the inverse of $-M$ has strictly positive entries: $(-M)^{-1} > 0$ elementwise. Also, stability implies that the determinant of $-M$ is positive: $\det(-M) > 0$. Then, $adj(-M) = (-M)^{-1}\det(-M) > 0$, hence the adjoint of $-M$ is also positive elementwise. To consider again the original sign of the variables, we change sign to the first row and column of $adj(-M)$, and we get the signature above for $adj(-J)$.

The next step is the characterization of the structural influence matrix, which corresponds to the sign pattern of the adjoint of the negative Jacobian matrix in *Proposition 1*.

To this aim, we first consider the linearized system and write it in a matrix-vector form

$$\dot{x}(t) = Jx(t) + e_j u$$

where $\dot{x}(t)$ is the time derivative of the four-dimensional vector $x(t)$ and $e_k$, $k = 1,2,3,4$, is an input vector, constant in time, with a single non-zero component, the $k$-th, equal to 1, while the scalar $u > 0$ is the magnitude of the input. We wish to assess the $i$-th component of $x(t)$, $x_i(t) = e_i^T x(t)$. If $J$ is Hurwitz, as assumed, the steady-state value of variable $x_i(t)$ due to the input perturbation $e_k$ applied to the equation of variable $x_k(t)$ is achieved for

$$0 = J\bar{x} + e_k u,$$

namely

$$x_i = -e_i^T J^{-1} e_k u,$$

which implies that the sign of the steady-state value $\bar{x}_i$ of variable $x_i$ due to a persistent positive input acting on the $k$-th equation has the same sign as $(-J^{-1})_{ik}$, the $(i,k)$ entry of matrix $(-J)^{-1}$. Since we assume Hurwitz stability, we have that $\det(-J)$ is positive, hence the sign pattern of the inverse $(-J)^{-1}$ corresponds to the sign pattern of the adjoint, $adj(-J)$. In fact, $adj(-J) = (-J)^{-1}\det(-J)$.

We next consider the nonlinear system under investigation, which we write in the form

$$\dot{x}(t) = f(x(t))$$

and without restriction we assume that the zero vector is an equilibrium point: $0 = f(0)$. This condition can be always achieved, without loss of generality, by a translation of coordinates. We also consider a stable equilibrium: we assume that the linearized system at the equilibrium is asymptotically stable, namely its Jacobian $J$, which has the sign pattern considered in *Proposition 1* above, is Hurwitz. We also assume that a constant input perturbation of magnitude $u$ is applied to the system, affecting the $k$-th equation, i.e.,

$$\dot{x}(t) = f(x(t)) + e_k u,$$

and that the perturbation is small enough to keep the state in the domain of attraction of the considered equilibrium. Due to this perturbation, a new steady state $\bar{x}(u)$ is reached that satisfies the condition

$$0 = f(\bar{x}(u)) + e_k u$$

To determine the sign of the new equilibrium components $\bar{x}(u)$, we consider this new equilibrium vector as a function of $u$ in a small interval $[0, x_{MAX}]$. Adopting the implicit function theorem yields

$$\frac{d}{dx}\bar{x}(u) = -J(u)^{-1} e_k u,$$

where we have denoted by $J(u)$ the Jacobian matrix computed at the perturbed equilibrium $\bar{x}(u)$. Hence, for $u$ small enough, the sign of the derivatives of the entries of the new, perturbed equilibrium are, structurally, the same as those in the $k$-th column of matrix $-J^{-1}$. Since, by construction, $x(0) = 0$, this is also the sign of the elements of vector $\bar{x}(u)$, for $u$ in the interval $[0, x_{MAX}]$.

We have therefore proved that the original nonlinear system describing honey bee health admits the following structural influence matrix:

$$\begin{bmatrix} + & - & - & - \\ - & + & + & + \\ - & + & + & + \\ - & + & + & + \end{bmatrix}$$

## System equilibria

The results concerning the system equilibria were obtained through a standard analytical treatment of the nonlinear equations describing the equilibrium conditions of the system of differential Eqs. (1), (2), (3), (4). A detailed description of methods is reported in Supplementary Methods.

## Laboratory experiments using honey bees

To confirm the bistability of the system representing honey bee health as affected by multiple stressors, we used data from several survival experiments, carried out in a laboratory environment according to the same standardized method, over a 6-year period (Source data file).

All experiments involved *Apis mellifera* worker bees, sampled at the larval stage or before eclosion, from the hives of the experimental apiary of the University of Udine (46°04′54.2″N, 13°12′34.2″E). Previous studies indicated that the local bee population consists of hybrids between *A. mellifera ligustica* and *A.m. carnica*[62,63]. Ethical approval was not required for this study.

We considered experiments on the effect of the following stressors: infection with 1000 DWV genome copies administered through the diet before pupation, feeding with a 50 ppm nicotine in a sugar solution at the adult stage, exposition to a sub-optimal temperature of 32 °C at the adult stage. All experiments were replicated 3 to 13 times, using, in total, the number of bees reported in Table 1.

For the artificial infection with DWV, we collected with soft forceps individual L4 larvae from the brood cells of several combs. Groups of 20–30 of such larvae were placed in Petri dishes with an artificial diet made of 50% royal jelly, 37% distilled water, 6% glucose, 6% fructose, and 1% yeast. 25 DWV copies per mg of diet were added or not to the diet according to the experimental group (note that a bee larva at this stage consumes about 40 mg of larval food per day, thus the viral infection per bee was 1000 viral copies). After 24 h larvae were transferred onto a piece of filter paper to remove the residues of the diet and then into a clean Petri dish, where they were maintained until eclosion. At the day of emergence, bees were transferred to plastic cages in a thermostatic cabinet, where they were kept until death. The DWV extract was prepared according to previously described protocols[64] and quantified according to standard methods.

For the treatment with nicotine, 10 μL of pure nicotine were added to 200 g of the sugar solution used for the feeding of the caged bees, to reach the concentration of 50 ppm.

Finally, to expose bees to a 32 °C temperature, the plastic cages with the adult bees were kept in a thermostatic cabinet whose temperature was set accordingly.

To monitor the survival of the adult bees treated as above, they were maintained from eclosion until death in plastic cages in a dark incubator at 34.5 °C (or 32 °C, according to the experiment), 75% R.H.; two syringes were used to supply a sugar solution made of 2.4 mol/L of glucose and fructose (61% and 31%, respectively) and water, respectively; dead bees were counted daily.

All the results of these experiments are reported in Source data file.

All experiments were carried out during the summer months, from June to September for 6 consecutive years. Previous data indicated that, in this region, virus prevalence increases along the active season starting from very low levels in spring and reaching 100% of virus-infected honey bees by the end of the summer; virus abundance in infected honey bees follows a similar trend[28]. For this reason, it can be assumed that bees sampled early in the season are either uninfected or they bear only a very low viral infection level, whereas bees sampled later in the season are likely to be virus-infected, bearing moderate to high viral infections. To confirm this assumption and identify a method for filtering our data according to viral infection, we assessed viral infection in a sample of bees from the untreated control group of each experiment, by means of qRT-PCR. According to standard practice, we assumed that Ct values below 30 are indicative of an effective viral infection, whereas Ct above that threshold are more likely in virus negative bees. As expected, we found that virus prevalence increases from June to September (Supplementary Figure 1a), in such a way that up to mid July only the minority of bees can be considered as viral infected (Supplementary Figure 1b). Therefore, we classified as "early" all the samples collected up to mid July and assumed that viral infection in those samples was low; on the other hand, samples collected from mid July till September were classified as "late" and we assumed that viral infection in those samples was high.

qRT-PCR analysis of viral infection was carried out as follows. At the beginning of every experiment (i.e., at day 0), two to five bees for each replication were sampled in liquid nitrogen and transferred in a −80 °C refrigerator. After defrosting of samples in RNA later, the gut of each honey bee was eliminated to avoid the clogging of the mini spin column used after. The whole body of sampled bees was homogenized using a TissueLyser (Qiagen®, Germany). Total RNA was extracted from each bee according to the procedure provided with the RNeasy Plus mini kit (Qiagen®, Germany). The amount of RNA in each sample was quantified with a NanoDrop® spectrophotomer (ThermoFisher™, USA). cDNA was synthetized starting from 500 ng of RNA following the manufacturer specifications (PROMEGA, Italy). Additional negative control samples containing no RT enzyme were included. DWV presence was verified by qRT-PCR considering as positive all samples with a Ct value lower than 30. The following primers were adopted: DWV (F: GGTAAGC-GATGGTTGTTTG, R: CCGTGAATATAGTGTGAGG[65]). 10 ng of cDNA from each sample were analyzed using SYBR®green dye (Ambion®) according to the manufacturer specifications, on a BioRad CFX96 Touch™ Real time PCR Detector. Primer efficiency was calculated according to the formula $E = 10^{(-1/\text{slope}-1)*100}$. The following thermal cycling profiles were adopted: one cycle at 95 °C for 10 min, 40 cycles at 95 °C for 15 s and 60 °C for 1 min, and one cycle at 68 °C for 7 min.

### Individual survival and colony stability

To investigate how the death rate of forager bees affects colony growth, a compartment model of honey bee colony population dynamics was proposed[50]. This model showed that death rates over a critical threshold led to colony failure. Here we modified this model to include premature death of bees at younger age, as predicted by our model of individual bee health in the presence of an immuno-suppressive virus. We show that the critical threshold found in the previously published model[50] becomes a decreasing function of the death rate of the younger individuals, so that premature death (and, in turn, immune-suppression) favors colony collapse.

In more details, we first summarize the results of the previously published model[50] where two populations $F$ (forager) and $H$ (hive) of bees are considered and where conditions are provided on the mortality $m$ of $F$ under which the whole population collapses: namely, mathematically stated, the system admits the zero equilibrium only. Here we extend the model partitioning $H$ in two categories, $Y$ (younger hive bees) and $O$ (older hive bees), as

$$H = Y + O$$

introducing an early mortality factor $n$ for the young population, showing how such a factor worsens the collapsing condition.

The previously published model[50] concerns the interaction between hive bees $H$ and forager bees $F$ and is described by the ODEs

$$\dot{H} = L\frac{H+F}{w+H+F} - H\left(\alpha - \sigma\frac{F}{H+F}\right)$$

$$\dot{F} = H\left(\alpha - \sigma\frac{F}{H+F}\right) - mF.$$

Above, $L$ is the queen's eggs laying rate, $w$ is the rate at which $L$ is reached as the total population $H + F$ gets large, $\alpha$ is the maximum rate at which hive bees become forager bees in the absence of the latter, $\sigma$ measures the reduction of recruitment of hive bees in the presence of forager bees and, finally, $m$ is the death rate of forager bees (while the death rate of hive bees is assumed to be negligible).

We first summarize the main results in terms of a threshold value for $m$ in view of colony collapse, as our further analysis will follow a similar approach. All the parameters are assumed to be positive.

The search for the equilibria of the above ODEs leads to the unique nontrivial equilibrium (beyond the trivial one)

$$\bar{H} = \frac{L}{mJ} - \frac{w}{1+J}$$

$$\bar{F} = J\bar{H}$$

for

$$J = J(m) : = \frac{\alpha - \sigma - m + \sqrt{(\alpha - \sigma - m)^2 + 4m\alpha}}{2m}.$$

Note that $J$ is alway positive (and, moreover, it is independent of $L$ and $w$). It follows that $\bar{F}$ and $\bar{H}$ have the same sign, so that the existence of the nontrivial equilibrium is equivalent to $\bar{F} + \bar{H} > 0$. It is not difficult to recover that

$$\bar{F} + \bar{H} = \frac{w}{m}\left(l\frac{1+J}{J} - m\right)$$

where $l : = L/w$ is introduced for brevity. Then if $\alpha \le l$ we get

$$\bar{F} + \bar{H} = \frac{w}{m}\left(l\frac{1+J}{J} - m\right) \ge \frac{w}{m}\left(\alpha\frac{1+J}{J} - m\right) = \frac{w}{m}(\sigma + mJ) > 0,$$

with the last equality following from

$$\alpha - \sigma\frac{J}{1+J} - mJ = 0,$$

which in turn comes from annihilating the right-hand side of the second ODE and from using $J = \bar{F}/\bar{H}$ while searching for equilibria. We conclude that, independently of $m$, the colony never collapses if the recruitment rate $\alpha$ of forager bees is sufficiently low.

Hence, we assume $\alpha > l$. Observe that

$$\bar{F} + \bar{H} \Longleftrightarrow l > J(m-l)$$

guarantees existence whenever $m$ is sufficiently small, viz. $m \le l$. Assume then $m > l$, so that the above condition reads

$$J < \frac{l}{m-l}$$

leading to the threshold condition

$$m < \bar{m} : = \frac{l}{2} \frac{\alpha + \sigma + \sqrt{(\alpha - \sigma)^2 + 4\sigma l}}{\alpha - l}$$

by using the definition of $J$, see Eq. (2) the previously published model[50].

A standard stability analysis shows that, assuming $\alpha, m > l$, the nontrivial equilibrium is (globally) asymptotically stable whenever it exists (positive), i.e., whenever $m < \bar{m}$. Otherwise, the only (globally) attracting equilibrium is the trivial one, corresponding to colony collapse (see Fig. 5 for the previously published model[50] or Fig. 4 for $n = 0$). In the mathematical jargon, the disappearance of the positive equilibrium, for $m$ exceeding $\bar{m}$, is referred to as a transcritical bifurcation[43].

Now, in view of the outcome of the analysis of our model of individual bee health, we introduce a mortality term for the younger bees. As forager bees are recruited from adult hive bees, we divide the class of hive bees $H$ in younger $Y$ and older $O$, assuming that the former die at a rate $n$, while the death rate of the latter remains negligible according to the previously published model[50]. Obviously, $H = Y + O$. The original ODEs are consequently modified as

$$\dot{Y} = L \frac{H+F}{w+H+F} - Y$$

$$\dot{O} = (1-n)Y - H\left(\alpha - \sigma \frac{F}{H+F}\right)$$

$$\dot{F} = H\left(\alpha - \sigma \frac{F}{H+F}\right) - mF.$$

Note that the sum of the first two equations above gives

$$\dot{H} = L \frac{H+F}{w+H+F} - H\left(\alpha - \sigma \frac{F}{H+F}\right) - nY.$$

The new negative mortality term for younger hive bees, $-nY$, models the fact that only the younger hive bees die prematurely while the rest of the dynamics is unchanged with respect to the original model.

The search for equilibria soon gives

$$\bar{Y} = L \frac{\bar{H} + \bar{F}}{w + \bar{H} + \bar{F}}$$

from the first ODE above, so that the remaining two equilibrium conditions lead to

$$\bar{H} = \frac{L_n}{mJ} - \frac{w}{1+J}$$

$$\bar{F} = J\bar{H}$$

for the same $J$ originally defined and $L_n : = L(1-n)$ (note that $n \in (0,1)$, and the case $n = 0$ brings us back to the original model). From this point on the analysis is the same as that previously summarized for the original model, but for replacing $L$ with $L_n$ and $l$ with $l : = l(1-n)$. Consequently, by assuming $\alpha, m > l_n$ (which is less restrictive when $n > 0$), the threshold condition $m < \bar{m}$ becomes

$$m < \bar{m}(n) : = \frac{l_n}{2} \frac{\alpha + \sigma + \sqrt{(\alpha - \sigma)^2 + 4\sigma l_n}}{\alpha - l_n},$$

which clearly returns the original threshold condition when $n = 0$. Since

$$\frac{d\bar{m}}{dn}(n) < 0$$

as it can be immediately verified, it follows that the critical value for $m$, $\bar{m}(n)$, beyond which the colony system admits only the zero equilibrium, i.e., the transcritical bifurcation value, decreases with $n$ (Fig. 4). We thus conclude that colony collapse is favored by the premature death of younger hive bees, possibly caused by a virus impairing the immune system as shown by the analysis of our model of individual bee health.

### Reporting summary

Further information on research design is available in the Nature Research Reporting Summary linked to this article.

### Data availability

The data generated in this study are provided in the Source data file. Source data are provided with this paper.

### Code availability

Figure 2 and Supplementary Figs. 2 and 3 were produced with custom codes developed with the software Mathematica (version 11.3.0.0 run on Mac OS X 10.11.6 MacBook Pro late 2013); Fig. 4 was produced with custom codes developed with the software MATLAB (version R2019a run on Mac OS X 11.6.1 MacBook Pro 2020). All the codes are freely available[66], also at: http://cdlab.uniud.it/software under the heading "BeeStability".

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

## Acknowledgements

European Union's Horizon 2020 research and innovation program, under grant agreement No. 773921 (PoshBee). C.J.T. was also funded by the European Union's Horizon 2020 research and innovation program, grant agreement No. 817622 (B-GOOD).

## Author contributions

Conceptualization: D.B., G.G., C.J.T., F.B., and F.N. Methodology: D.B., G.G., D.A., F.B., and F.N. Investigation: D.B., D.F., G.G., E.S., V.Z., D.A., F.B., F.N. Visualization: D.B., D.A. Supervision: F.B. and F.N. Writing—original draft: D.B., G.G., D.A., C.J.T., F.B., and F.N. Writing—review & editing: D.B., D.F., G.G., E.S., V.Z., D.A., C.J.T., F.B., and F.N.

## Competing interests

The authors declare no competing interests.
