## [Peer Review File · Nature Communications]

Reviewers' Comments:

Reviewer #1:

Remarks to the Author:

In this manuscript, the authors aim at understanding the processes that can explain discrepancies in findings between different field studies or field studies and lab studies on the effects of pesticides, particularly neonicotinoids, on honeybees. I believe this question is of interest to a sufficiently broad scientific audience to be considered for publication in Nature communications. However, I have my doubts that in its current form, the target audience, to which I belong as an ecologist / environmental scientist researching bees, understands easily enough what the authors did and what the results mean. I must admit that to me even the core methods are still unclear after reading the whole manuscript including the supplementary material and I think this should be clear without even reading the methods section.

Nonetheless, I will summarize my understanding of the manuscript (which may be wrong). The authors used a systems biology approach to better understand how the multitude of stressors can impact the results obtained when focusing on a single stressor (particularly pesticides). For this, they built a conceptual model of factors potentially impacting honeybee health including indirect and interactive effects. Based on this conceptual model they built a set of differential equations describing honeybee health and the stress due to viruses (or a virus: DWV), parasites (or a parasite: varroa) and toxins (or a class of pesticides: neonicotinoids). Then they ran a structural analysis that consists in searching system equilibria by equating honeybee health in the equation on the stressor with honeybee health in the equation on honeybee health itself. For this they make several assumptions e.g. that the effects of stressors can only be negative but not so strong that no survival is possible. They find that a single equilibrium exists when looking at the subsystems of honeybee health and either toxins alone or parasites alone or toxins and parasites but not when looking at the subsystems of honeybee health and viruses, if and only if these are immune-suppressive viruses. In the latter case, they find three equilibria, which they claim reconciles discrepancies in the findings of different field studies on insecticide effects on bees. They claim they validated the conceptual/mathematical model by using 5 years of data from lab studies investigating effects of seasonality (early vs late-season), DWV infection, nicotine exposure and/or temperature on honeybee longevity. In addition, they use a mathematical model to determine the effects of reduced lifespan on colony stability.

The authors refer in the title to "bees" even though they try to explain contrasting results in honeybees only. Generally, they could make a clearer distinction when they mean honeybees and when they mean other bees or bees in general.

They use the term "interaction" as it is apparently done in systems biology, which is fine, but I suggest that they clarify this as it is different from how this term is typically used by statisticians and ecologists), where "interaction" (or "interactive effects") refers to the fact that the effect of one factor on a response variable depends on the effect of another factor. The conceptual model that they built contains a few interactive effects and one positive feedback loop. For the one positive feedback loop they include, DWV inhibits immune system, which fosters DWV proliferation, they find different system equilibria. The authors did not include interactive effects between different pesticides even though it has been repeatedly shown that fungicides can increase the toxicity of insecticides probably by inhibiting detoxification (McArt et al., 2017; Pilling and Jepson, 1993; Sgolastra et al., 2017; Wernecke et al., 2019).

The conceptual model does not include factors that may result in different pesticide exposure, which in field studies is a crucial point and is therefore relevant if the goal is to explain discrepancies between different field studies. For instance, low temperature is included as a positive driver of bee health as it may increase nectar feeding, but increased nectar feeding may also result in increased pesticide intake. In addition, flowering resources can impact the way pesticide use affects bees either by deterring them from treated plants or potentially also by altering their pesticide tolerance (Klaus et al., 2021; Park et al., 2015; Vaudo et al., 2015; Wintermantel et al., 2022). On a side note, it is possible that pesticide use has also indirect positive impacts on bees through increasing flower density or nectar flow.

I think a clearer description of what the system equilibria and the orbits in Fig. 2 are is needed. I presume the orbits are the values that the honeybee health index can take for different values of the explanatory variables, but I cannot figure out how this relates to the system equilibria.

The authors conducted four types of lab experiments if I see that correctly (even though they write of a "second experiment" but not of a third or a fourth experiment, which I find slightly confusing. In all of the experiments, they caged individual honeybees and measured longevity of individuals. In the first type, bees were caught at different times of the year and not exposed to any additional stressor. Longevity was here related to the date they were sampled whereby date was a categorical variable (early season, late season). The authors claim that they find a unimodal longevity distribution for early-season honeybees and a bimodal longevity distribution for late-season honeybees. However, by looking on Fig. 3A I cannot see this. It looks to me like in both cases there is a slight peak for bees that die within 4 days and then another more pronounced one much later (after 24 to 28 days).

In a second type of experiment, they additionally administered a defined amount of DWV to a subset of the honeybees and claim their results support findings from the previous experiment. And that this would support the view that the "presence of an immune-suppressing virus can determine vastly different outcome depending on the slightly different initial conditions of single bees exposed to otherwise identical situations". I am not sure what is meant by "slightly different initial conditions", it refer to the time of the years that the bees were sampled but this is not shown in the Figure 3B that the authors refer to or refer to the viruses that were either administered or not, which would be a complicated way of saying that artificial DWV infection negatively affected honeybee longevity. Although I do see that in this case, it is more apparent that not only the mean was shifted but the distribution as well, I don't see a shift from unimodal to bimodal distribution.

In a third type of lab experiment, early-season and late-season honeybees were exposed to nicotine or not (control) and in a fourth one early-season and late-season honeybees were exposed to one of two different temperatures. In the latter, 32 °C was chosen as the suboptimal temperature but I believe in the manuscript it is not mentioned what is chosen as the optimal or reference temperature.

As far as I can see, the authors did not conduct any statistical analyses but simply calculated standard deviations and medians, which is sort of incoherent as standard deviations are really only meaningful when data are normally distributed and median instead of mean values are typically reported when data are not normally distributed, which seems generally to be the case as the histograms indicate. Also, the categorization in early-season and late-season bees is quite crude, which I do not find per se problematic as a simplification but I would like to see if the trends that are observed are the same if date is used as a continuous variable. Perhaps degree days could be used if data from multiple years are used to account for differences in the start of the beekeeping season. Generally, it is unclear to me if each type of experiment was conducted in a separate year or if some experiments spanned over several years or were repeated. Generally, I find that they have interesting lab experiments that are however not given much attention. These results are presented as simple validation of the model, which sort of undermines that their findings are relevant in and of by themselves. However, it is up to the authors where to put the focus and I may just focus too much on the aspects that I am familiar with.

In the 3-dimensional histograms in Fig. 3C and Fig. 3D, some bars cannot be seen. They nonetheless allow to see the described patterns but I advise the authors to overlay different density plots to circumvent this problem. If it becomes too difficult to distinguish the density plots various facets may be used (where each shows e.g. two density plots).

I don't understand why in Fig. 4 colony population sizes below zero are illustrated. I must acknowledge that I did not review the Figures in the Supplementary Material in detail.

Finally, I will challenge the authors' claim that this study explains the contrasting results of neonicotinoid effects between different studies. They claim that negative effects of neonicotinoids are only observable when DWV (or the Varroa mite as a proxy of DWV loads) is prevalent.

However, they also claim that “negative effects of neonicotinoid insecticides have been clearly established in the laboratory, field testing has resulted in contradictory outcomes”. There is no reason to believe that in lab studies honeybees have consistently higher DWV levels than in field studies. Besides, Woodcock et al.’s results are inaccurately reported in lines 269-271. Unlike stated by the authors, in Hungary where negative effects were observed, Varroa mite levels were (almost) equally low as in Germany where positive effects were observed. Also, the authors cite Osterman et al. (2019) as an example of a study where DWV prevalence and/or mite infestation was low that showed no adverse neonicotinoid effects. However, Osterman et al. reported that at the start of the experiment in 2013, around three-quarters of the colonies were infected with DWV Type-2. In addition, studies on the interactive effects of viruses (including DWV) and pesticides on bees have come to equally contrasting results (Harwood and Dolezal, 2020) as studies assessing only effects of neonicotinoids. In fact, studies finding synergistic effects are a minority. Besides, the study by di Prisco et al. that the authors of this article cite used primers that were criticized for producing a large number of PCR artefacts dominating the quantitative signal (Osterman et al., 2019). I do not want to say that DWV (or other pathogens) and neonicotinoids (or other pesticides) cannot interact but that the evidence that DWV drives differences in observed neonicotinoid effects in different (field) studies is slim.

For the above-mentioned reasons, I do not recommend acceptance of the manuscript in its current form even though the authors address a relevant question which they tackle from various aspects, which I find an interesting approach. I do not feel competent to judge whether or not the manuscript can be revised so that it reaches the required standard for Nature communications as my understanding of a core part of the study is limited.

Further comments can be found below:

L40-41: The study by Neumann et al. is already 12 years old. To get a more up-to-date and holistic overview of honeybee colony losses, I recommend the authors to read Osterman et al., 2021 who systematically reviewed reported overwintering mortality and found considerable regional differences but no consistent increase over time.

L51: I suggest the authors provide at least one reference showing that “negative effects of neonicotinoid insecticides have been clearly established in the laboratory”.

L244: I suggest the authors cite here also a paper that reviews impacts of neonicotinoids on bees rather than only three original research articles that are (almost) 10 years old. There are plenty of reviews available and also a few meta-analyses.

Table S1: For the Swedish field studies the authors forgot to mention that *Osmia bicornis* was assessed too (Rundlöf et al., 2015). Also, they summarize the findings as follows: “No impact on honey-bees, effects on bumble bees”. It would be more accurate and consistent with the reporting of the results of the other field studies to state that there were negative effects on bumblebees and red mason bees. Also, in the same field experiment some positive effects on honeybees were found (Osterman et al., 2019).

References

Harwood, G.P., Dolezal, A.G., 2020. Pesticide–Virus Interactions in Honey Bees: Challenges and Opportunities for Understanding Drivers of Bee Declines. *Viruses* 12, 566. <https://doi.org/10.3390/v12050566>

Klaus, F., Tschardtke, T., Bischoff, G., Grass, I., 2021. Floral resource diversification promotes solitary bee reproduction and may offset insecticide effects – evidence from a semi-field experiment. *Ecol. Lett.* 24, 668–675. <https://doi.org/10.1111/ele.13683>

McArt, S.H., Fersch, A.A., Milano, N.J., Truitt, L.L., Böröczky, K., 2017. High pesticide risk to honey bees despite low focal crop pollen collection during pollination of a mass blooming crop. *Sci. Rep.* 7, 1–10. <https://doi.org/10.1038/srep46554>

Osterman, J., Aizen, M.A., Biesmeijer, J.C., Bosch, J., Howlett, B.G., Inouye, D.W., Jung, C.,

Martins, D.J., Medel, R., Pauw, A., Seymour, C.L., Paxton, R.J., 2021. Global trends in the number and diversity of managed pollinator species. *Agric. Ecosyst. Environ.* 322, 107653. <https://doi.org/10.1016/j.agee.2021.107653>

Osterman, J., Wintermantel, D., Locke, B., Jonsson, O., Semberg, E., Onorati, P., Forsgren, E., Rosenkranz, P., Rahbek-Pedersen, T., Bommarco, R., Smith, H.G.H.G., Rundlöf, M., de Miranda, J.R.J.R., 2019. Clothianidin seed-treatment has no detectable negative impact on honeybee colonies and their pathogens. *Nat. Commun.* 10, 1–13. <https://doi.org/10.1038/s41467-019-08523-4>

Park, M.G., Blitzer, E.J., Gibbs, J., Losey, J.E., Danforth, B.N., 2015. Negative effects of pesticides on wild bee communities can be buffered by landscape context. *Proc. R. Soc. B Biol. Sci.* 282. <https://doi.org/10.1098/rspb.2015.0299>

Pilling, E.D., Jepson, P.C., 1993. Synergism between EBI fungicides and a pyrethroid insecticide in the honeybee (*Apis mellifera*). *Pestic. Sci.* 39, 293–297. <https://doi.org/10.1002/ps.2780390407>

Rundlöf, M., Andersson, G.K.S., Bommarco, R., Fries, I., Hederström, V., Herbertsson, L., Jonsson, O., Klatt, B.K., Pedersen, T.R., Yourstone, J., Smith, H.G., 2015. Seed coating with a neonicotinoid insecticide negatively affects wild bees. *Nature* 521, 77–80. <https://doi.org/10.1038/nature14420>

Sgolastra, F., Medrzycki, P., Bortolotti, L., Renzi, M.T., Tosi, S., Bogo, G., Teper, D., Porrini, C., Molowny-Horas, R., Bosch, J., 2017. Synergistic mortality between a neonicotinoid insecticide and an ergosterol-biosynthesis-inhibiting fungicide in three bee species. *Pest Manag. Sci.* 73, 1236–1243. <https://doi.org/10.1002/ps.4449>

Vaudo, A.D., Tooker, J.F., Grozinger, C.M., Patch, H.M., 2015. Bee nutrition and floral resource restoration. *Curr. Opin. Insect Sci.* 10, 133–141. <https://doi.org/10.1016/j.cois.2015.05.008>

Wernecke, A., Frommberger, M., Forster, R., Pistorius, J., 2019. Lethal effects of various tank mixtures including insecticides, fungicides and fertilizers on honey bees under laboratory, semi-field and field conditions. *J. fur Verbraucherschutz und Leb.* 14, 239–249. <https://doi.org/10.1007/s00003-019-01233-5>

Wintermantel, D., Pereira-Peixoto, M.-H., Warth, N., Melcher, K., Faller, M., Feuerer, J., Allan, M.J., Dean, R., Tamburini, G., Knauer, A.C., Schwarz, J.M., Albrecht, M., Klein, A.-M., 2022. Flowering resources modulate the sensitivity of bumblebees to a common fungicide. *Sci. Total Environ.* 829, 154450. <https://doi.org/10.1016/j.scitotenv.2022.154450>

Reviewer #2:

Remarks to the Author:

The paper relates experimental results obtained on honey bees colonies with a qualitative analysis of mathematical models that describe the main interactions.

I am not expert in the specifics of honey bees ecology, so my review will only be addressing the mathematical aspects and qualitative results proposed.

The manuscript appears to be technically sound, to the best of my knowledge, and provides an interesting instance of qualitative analysis tools enabling interpretation and reconciliation of apparently contradicting empirical observations.

I have some minor suggestions for improving clarity of presentation:

Line 109-110: "by different functions which are all decreasing to 0 because they exert a negative effect on honey bee health".

This sentence is slightly misleading as it stands.

I believe a better formulation would be: "by different monotonically decreasing functions because each factor exerts a negative effect on honey bee health."

The fact that these are decreasing to zero is perhaps a suitable modelling assumption but is unrelated to the fact that the effect is negative.

Line 123: "if the sign of the first state variable is changed."

While I understood what authors meant I believe the sentence is unclear as it seems to imply an arbitrary manipulation of the model that may alter its significance or dynamical behaviour.

Perhaps the following formulation improves readability:

"If the model is reformulated by using as a first state variable the opposite of x_{BH} (viz. an indicator of bee unhealthiness)"

Line 364: It seems to me that matrix M is Hurwitz by assumption (as is similar to J), not because of its sign pattern.

Overall, while I found the phenomena modelled and described in the paper to be a convincing paradigm of a situation in which qualitative analysis is important to interpret and validate field studies.

REVIEWER COMMENTS

Reviewer #1 (Remarks to the Author):

In this manuscript, the authors aim at understanding the processes that can explain discrepancies in findings between different field studies or field studies and lab studies on the effects of pesticides, particularly neonicotinoids, on honeybees. I believe this question is of interest to a sufficiently broad scientific audience to be considered for publication in Nature communications. However, I have my doubts that in its current form, the target audience, to which I belong as an ecologist / environmental scientist researching bees, understands easily enough what the authors did and what the results mean. I must admit that to me even the core methods are still unclear after reading the whole manuscript including the supplementary material and I think this should be clear without even reading the methods section.

We are glad that the Reviewer agrees with us on the importance of the observed discrepancies between different studies on the effects of neonicotinoids on honeybees.

We understand the problems the Reviewer has experienced in trying to understand the methods we used; actually, this is the same problem that the bee biologists in our research team faced when confronting the mathematical analysis carried out by our colleagues working in systems biology.

In general, this work made all of us aware of a general and quite important point: some open problems require an interdisciplinary effort, but each discipline has got methods, jargon and styles that are peculiar. Nevertheless, we aimed to write a report that could be understandable by everybody regardless of his or her background; the comments by the Reviewer indicate that we have not yet reached that level of clarity.

Below we describe the actions we took to make our paper clear for the largest possible audience, without sacrificing precision and detail.

Nonetheless, I will summarize my understanding of the manuscript (which may be wrong). The authors used a systems biology approach to better understand how the multitude of stressors can impact the results obtained when focusing on a single stressor (particularly pesticides). For this, they built a conceptual model of factors potentially impacting honeybee health including indirect and interactive effects. Based on this conceptual model they built a set of differential equations describing honeybee health and the stress due to viruses (or a virus: DWV), parasites (or a parasite: varroa) and toxins (or a class of pesticides: neonicotinoids). Then they ran a structural analysis that consists in searching system equilibria by equating honeybee health in the equation on the stressor with honeybee health in the equation on honeybee health itself. For this they make several assumptions e.g. that the effects of stressors can only be negative but not so strong that no survival is possible. They find that a single equilibrium exists when looking at the subsystems of honeybee health and either toxins alone or parasites alone or toxins and parasites but not when looking at the subsystems of honeybee health and viruses, if and only if these are immune-suppressive viruses. In the latter case, they find three equilibria, which they claim reconciles discrepancies in the findings of different field studies on insecticide effects on bees. They claim they validated the conceptual/mathematical model by using 5 years of data from lab studies investigating effects of seasonality (early vs late-season), DWV infection, nicotine exposure and/or temperature on honeybee longevity. In addition, they use a mathematical model to determine the effects of reduced lifespan on colony stability.

We confirm that the Reviewer understood the fundamental meaning of our work.

The authors refer in the title to “bees” even though they try to explain contrasting results in honeybees only. Generally, they could make a clearer distinction when they mean honeybees and when they mean other bees or bees in general.

We thank the Reviewer for pointing this aspect to our attention. While the approach we undertook could certainly be applied to other bees, the conceptual model we started from is based on honey bee biology and therefore conclusions should be restricted to this biological system. We have now clarified in the title, and whenever necessary, in the text, that we deal with honey bees only, and not with bees in general, so as to prevent any possible misunderstanding.

They use the term “interaction” as it is apparently done in systems biology, which is fine, but I suggest that they clarify this as it is different from how this term is typically used by statisticians and ecologists), where “interaction” (or “interactive effects”) refers to the fact that the effect of one factor on a response variable depends on the effect of another factor. The conceptual model that they built contains a few interactive effects and one positive feedback loop. For the one positive feedback loop they include, DWV inhibits immune system, which fosters DWV proliferation, they find different system equilibria. The authors did not include interactive effects between different pesticides even though it has been repeatedly shown that fungicides can increase the toxicity of insecticides probably by inhibiting detoxification (McArt et al., 2017; Pilling and Jepson, 1993; Sgolastra et al., 2017; Wernecke et al., 2019).

As the Reviewer correctly understood, we used the term “interaction” in its general form and do not refer to statistical interactions here. We have now added a few words to clarify this aspect (see line 75).

However, the point about the interactive effects of pesticides allows us to clarify another important aspect. In our conceptual model we included one single toxic compound, despite many pesticides, interacting with each other in a statistical fashion, can impact honey bees at the same time. This may seem an oversimplification of the system which in turn could affect the significance of our conclusions. This would certainly be the case if ours were a descriptive model aiming at quantifying bee health at any given time, in presence of a defined amount of certain stressors. However, ours is not that kind of model and, for the purpose of our structural analysis, one toxic compound exerting a negative effect, or two toxic compounds interacting with each other, so as to exert an even bigger negative effect, does not make a difference as far as the sign of the resulting effect is the same. With reference to Fig. 1A, as far as our analysis is concerned, there is no difference between one single square named toxic compounds connected with a broken arrow to honey bee health and many little squares (one for each toxic compounds that bees can encounter in the environment), connected to each other by pointed arrows (accounting for the reciprocal interactions) and to honey bee health by broken arrows, such that, at the end, we can embed all those little squares into a bigger one, connected to honey bee health by one single broken arrow accounting for the resulting interactive effect of the many pesticides.

Knowing the complexity of the system we deal with, and the great deal of work that has recently been done on interactive effects by a number of colleagues (and ourselves), we are well aware of this problem and therefore tried to clarify in advance that by no means we wanted to factor out this complexity from our analysis. We therefore added a dedicated paragraph after the presentation of our conceptual model, which apparently was not clear enough. Now, we have

extended that paragraph to make sure that this central concept of our study is clearer (see lines 103-121).

The conceptual model does not include factors that may result in different pesticide exposure, which in field studies is a crucial point and is therefore relevant if the goal is to explain discrepancies between different field studies. For instance, low temperature is included as a positive driver of bee health as it may increase nectar feeding, but increased nectar feeding may also result in increased pesticide intake.

We fully agree with the Reviewer on this point and already included this possible effect in our analysis. In fact, in Fig. 1A there is a green arrow (“l”) going from “low temperature” to “quantity of sugar supply”, to indicate that, indeed, low temperature may increase nectar feeding. The component “quantity of sugar supply” is then connected to “quality of sugar supply” which receives a red arrow (“h”) from toxic compounds, meaning that, as the Reviewer correctly noted, increased nectar feeding may result in increased pesticide intake.

In mathematical terms, this is captured by the function $f_{S,C}(u_s, u_C, x_{TC}, x_{VA})$ in the first differential equation of our model. The fact that this function is increasing with respect to u arguments and in particular with respect to u_s , representing nectar feeding, means that the more nectar the better bee health. However, the fact that the function is decreasing with respect to x arguments, and in particular with respect to x_{TC} , representing toxic compounds, implies that the “positive” contribution of nectar feeding can be negatively modulated by toxic compounds, as the Reviewer correctly pointed out.

We have highlighted this concept in the manuscript (see lines 96 and 130-135).

In addition, flowering resources can impact the way pesticide use affects bees either by deterring them from treated plants or potentially also by altering their pesticide tolerance (Klaus et al., 2021; Park et al., 2015; Vaudo et al., 2015; Wintermantel et al., 2022).

Again, we agree with the Reviewer that “clean” floral resources may compete with treated plants resulting in a lower input of pesticides; however, this hasn’t got any implication for our model since it would simply correspond to a reduced negative impact of toxic compounds, which in fact can assume any value from zero to infinity.

Also, we agree that compounds from plants could alter pesticide tolerance; in particular, some compounds could increase tolerance and thus positively influence bee health, but this effect is already encapsulated in arrow “e” exiting from “nectar” and entering bee health; on the other hand, other compounds from plants could decrease pesticide tolerance, but this would simply correspond to a higher input of arrow “g” coming from toxic compounds. In other words, the effects mentioned by the Reviewer are certainly biologically relevant but do not affect the structure of our model and thus would not alter the results of our analysis. Nevertheless, we agree they are worth mentioning; similarly, it is worth explaining why those effects would not alter the behavior of the system and our conclusions. In the revised text, we have included a clarification (see lines 117-121).

On a site note, it is possible that pesticide use has also indirect positive impacts on bees through increasing flower density or nectar flow.

As mentioned above we have now better clarified our approach.

I think a clearer description of what the system equilibria and the orbits in Fig. 2 are is needed. I presume the orbits are the values that the honeybee health index can take for different values of the explanatory variables, but I cannot figure out how this relates to the system equilibria.

In the revised text we better explained both what the orbits represent and what the equilibria are (see lines 169-177).

The authors conducted four types of lab experiments if I see that correctly (even though they write of a “second experiment” but not of a third or a fourth experiment, which I find slightly confusing.

We carried out four types of lab experiment; in the revised text we fixed this problem (see line 274).

In all of the experiments, they caged individual honeybees and measured longevity of individuals. In the first type, bees were caught at different times of the year and not exposed to any additional stressor. Longevity was here related to the date they were sampled whereby date was a categorical variable (early season, late season). The authors claim that they find a unimodal longevity distribution for early-season honeybees and a bimodal longevity distribution for late-season honeybees. However, by looking on Fig. 3A I cannot see this. It looks to me like in both cases there is a slight peak for bees that die within 4 days and then another more pronounced one much later (after 24 to 28 days).

In our manuscript we preferred a graphical representation of the observed effect not to obscure our principal finding with too many details. In fact, a closer look at Fig. 3A reveals that the distribution of longevity data in the case of early season bees differs from the corresponding distribution of late season bees. In the first case, a very high green bar can be noted in the interval 20-24 days of age corresponding to 50% of all bees with a longevity between 21 and 24 days of age (please note that this is largely expected in the case of an insect normally showing a type I survival curve under lab conditions).

Instead, if you look at the orange bars you may notice that there is not such a concentrated mode (i.e. only 20% of bees had longevities between 21 and 24 days), while a large number of individuals (another 20%) actually died before reaching 12 days of age, as highlighted by the three orange bars in the left part of the graph (this proportion is 4 % in the case of early season bees).

Indeed, if no “early season” bees had died before let’s say 12 days, our results would have probably looked more convincing but, under our hypothesis, this would be plausible only if there were no virus infected bees, corresponding to a 0% virus prevalence; unfortunately, this was not the case. In fact, as illustrated in table 1, virus prevalence was 0.09 early in the season, meaning that one out ten bees was actually virus infected and could undergo the same destiny as the 7 out of ten bees in the “late season” group.

We believe that results are consistent with our predictions; to better explain this central concept, we revised the text and commented more extensively the distribution of longevities (see lines 272-282).

In a second type of experiment, they additionally administered a defined amount of DWV to a subset of the honeybees and claim their results support findings from the previous experiment. And that this would support the view that the “presence of an immune-suppressing virus can determine

vastly different outcome depending on the slightly different initial conditions of single bees exposed to otherwise identical situations". I am not sure what is meant by "slightly different initial conditions", it refer to the time of the years that the bees were sampled but this is not shown in the Figure 3B that the authors refer to or refer to the viruses that were either administered or not, which would be a complicated way of saying that artificial DWV infection negatively affected honeybee longevity. Although I do see that in this case, it is more apparent that not only the mean was shifted but the distribution as well, I don't see a shift from unimodal to bimodal distribution.

To test the effect of an immune-suppressing virus on the distribution of longevities of honey bees, we needed to compare two groups of bees, infected and uninfected with the virus.

To double check our hypothesis we carried out two different kinds of comparisons. In the first experiment the comparison between virus infected and virus free bees was diachronic (i.e. no treatment but just different conditions at different times), exploiting the circumstance that virus prevalence increases along the season. Instead, in the second experiment, we carried out a synchronic comparison (i.e. one treatment to cause different conditions at the same time), by treating or not with the virus bees that were expected to be uninfected.

When we speak about "vastly different outcome depending on the slightly different initial conditions of single bees exposed to otherwise identical situations" we mean that the destiny of uninfected bees simply depends on the treatment they receive, while the destiny of virus infected bees is very sensitive with respect to minor differences that are difficult to control. As a results of that, uninfected bees tend to have a similar longevity but virus infected bees can have either long longevities or shorter longevities producing what we called a bimodal distribution. In other words, the "slightly different conditions" are not the time of the year (e.g. early vs late) or the administration of the virus (e.g. virus yes vs virus no), those are just the treatments; the "slightly different conditions" are the intrinsic individual conditions of the virus infected bees, which were not under our control, and determined their destiny.

This is a crucial concept; therefore, in the revised version of the manuscript we tried to express it more clearly (see lines 263-272); thank you for providing your illuminating point of view.

In a third type of lab experiment, early-season and late-season honeybees were exposed to nicotine or not (control) and in a fourth one early-season and late-season honeybees were exposed to one of two different temperatures. In the latter, 32 °C was chosen as the suboptimal temperature but I believe in the manuscript it is not mentioned what is chosen as the optimal or reference temperature.

In the methods, in the section about the "Lab studies on caged bees" we wrote: "To monitor the survival of the adult bees treated as above, they were maintained in plastic cages in an incubator at 34.5 °C". However, to make this data more visible we have now added this information in the main text as well (see lines 276-277).

As far as I can see, the authors did not conduct any statistical analyses but simply calculated standard deviations and medians, which is sort of incoherent as standard deviations are really only meaningful when data are normally distributed and median instead of mean values are typically reported when data are not normally distributed, which seems generally to be the case as the histograms indicate.

It is correct that we did not conduct any statistical analysis and just used the standard deviation to represent the variability of longevity data. We did so despite the not-normal distribution of our data because, in principle, standard deviation can be regarded as a useful algorithm to represent how dispersed a set of data is. In addition, when data are normally distributed, a certain proportion of data lies within one or two standard deviations from the sample mean so that, in this case, this parameter can be used for comparative purposes, to assess how significant is the distance between the means to two samples. However, in our case, we were not interested in comparing means and did not use the standard deviation for that purpose.

In any case, the use of standard deviation for the purpose of comparisons is very common and indeed our choice could be misleading. Therefore, following the Reviewer's request, we now use another measure of dispersion, the interquartile range (see lines 251-252 and following). As expected, this did not affect the conclusions we drew using standard deviation to measure data variability.

Also, the categorization in early-season and late-season bees is quite crude, which I do not find per se problematic as a simplification but I would like to see if the trends that are observed are the same if date is used as a continuous variable. Perhaps degree days could be used if data from multiple years are used to account for differences in the start of the beekeeping season.

We fully agree with the Reviewer that the categorization "early" vs "late" is crude and a preliminary look at data using time as a continuous variable would be advisable. In fact, in Fig. S1A we report the prevalence of DWV in our bees according to the month and show that our choice, albeit crude, is supported by available data, in that virus prevalence gradually increases along the season. We admit that, also in this case, time was treated as a discrete variable (i.e. we used 4 months, instead of 2 periods, and this is still worse than 120 days); however, to have more data points, one would need to assess viral infection on a much larger number of bees at many more time points which is something we did not consider as essential in view of available literature and our own experimental data reported here, which confirmed previous results.

In any case we now cite Fig. S1 in the main text where we first mention early and late-season bees (see line 243) to show how our classification was supported.

Generally, it is unclear to me if each type of experiment was conducted in a separate year or if some experiments spanned over several years or were repeated.

In general, we refer to experiments carried out in our lab in a rather long period of time, which we specified in the methods. However, for more details the reader is referred to the excel file where all raw data are reported including the exact date of each single experiment.

In any case, we have now added a few lines to the results section of the manuscript where we clarify this circumstance (see lines 235-238).

Generally, I find that they have interesting lab experiments that are however not given much attention. These results are presented as simple validation of the model, which sort of undermines that their findings are relevant in and of by themselves. However, it is up to the authors where to put the focus and I may just focus too much on the aspects that I am familiar with.

We thank the Reviewer for appreciating the quality of the data we are presenting here. We agree that they are precious data; in fact, we plan to use them for further studies dealing with different research questions. However, as it has been noted, here they are used for validating our new

model which, in our opinion, is quite an important scope, providing a deeper understanding of the phenomenon under study based on a solid mathematical language.

In the 3-dimensional histograms in Fig. 3C and Fig. 3D, some bars cannot be seen. They nonetheless allow to see the described patterns but I advise the authors to overlay different density plots to circumvent this problem. If it becomes too difficult to distinguish the density plots various facets may be used (where each shows e.g. two density plots).

We thank the Reviewer for this suggestion.

We agree that by overlaying different density plots, no bars would be hidden beyond any other and, moreover, results may look better due to the smoothing of curves.

For example, the figure below shows the density plots of early and late control bees reported in Fig. 3A.

However, for presenting graphs like this we would need to make at least three arbitrary assumptions: one regarding bandwidth (e.g., 1 in the graph above), another regarding the number of intervals (e.g., 50 in the graph above) and a further one regarding the kernel distribution (e.g., normal distribution in graph above). All those decisions really affect the palatability of results. See for example what happens if bandwidth is 0.5 and the number of intervals is restricted to 30.

Actually, to display our graphs we have already arbitrarily selected the width of the interval (although in our supplementary excel file anyone can change this assumption and see what the graphs look like), but we prefer not to make any further unnecessary assumption and, moreover, we do not wish to give any misleading impression concerning samples size and/or quality of data distribution.

For this reason, since, as the Reviewer noted, our histograms allow to see the described patterns, we would keep the graphical representation we first proposed.

In any case, we provide all the raw data which we based our work on in a supplementary excel file, including the formulas that can be used to automatically generate the data sets used to produce graphs, so that any reader can visualize the data as she/he prefers.

I don't understand why in Fig. 4 colony population sizes below zero are illustrated. I must acknowledge that I did not review the Figures in the Supplementary Material in detail.

This graph is a bifurcation diagram representing the colony population at the equilibrium for varying death rates “m” of juvenile hive bees. In practice, along the lines one can read the number of bees in the colony after the equilibrium is reached, if the death rates correspond to the value represented on the x axis. For example, if m is 0.355 then population at equilibrium is 0, meaning that m cannot be bigger than that value. In this kind of graphs, stable values are normally represented with solid lines and unstable values with dotted lines, and this is what we did to facilitate graph interpretation.

We used this graph because this is the same display used, in their article (see ref. 45 in the manuscript), by the authors of the model we modified in this paper to see the effect of an extra-mortality occurring at a younger age on colony sustainability.

Of course, values below zero are mathematical meaningful (for the sake of bifurcation) yet biologically meaningless. In other words, as the Reviewer correctly noticed, in a real bee colony, total population cannot go below zero, but according to the equation reported in the methods, the variable $Y+O+F$ could reach negative values, and this is why those values are reported with dashed lines (to warn the reader that they are unfeasible values). The main purpose of this graph, however, is not to follow the “behavior” of the variable “colony population” but rather to find what are the values of “m” that are consistent with colony sustainability.

We have now revised the text so that both the purpose and the conclusions of this part of the study can be clearer (see lines 297-304).

Finally, I will challenge the authors' claim that this study explains the contrasting results of neonicotinoid effects between different studies. They claim that negative effects of neonicotinoids are only observable when DWV (or the Varroa mite as a proxy of DWV loads) is prevalent. However, they also claim that "negative effects of neonicotinoid insecticides have been clearly established in the laboratory, field testing has resulted in contradictory outcomes". There is no reason to believe that in lab studies honeybees have consistently higher DWV levels than in field studies.

We are not suggesting that the negative effects of neonicotinoids are only observable when DWV (or the Varroa mite as a proxy of DWV loads) is prevalent.

Instead, according to our analysis, we propose that:

- a. if a certain stressor is above a certain level there is only one equilibrium at low bee health; meaning that, for example, if a neonicotinoid insecticide is present at high concentration, bee survival will be significantly shorter than normal, and a negative effect will be noted;
- b. if the same stressor is below that level, one equilibrium at high bee health is certainly possible; meaning that for example, if the neonicotinoid insecticide is present at low concentration, bee survival may be not different from normal and a negative effect may not be noted;
- c. in the presence of an immune suppressing virus, multistability can occur; so that for the same intermediate level of one stressor, one can have either low bee health or high bee health depending on the initial conditions, and results may become unpredictable. In other words, in the presence of an intermediate amount of insecticide, each virus infected bee can either die prematurely or survive much longer depending on its intrinsic individual situation at the beginning, and this can then propagate at colony level.

Also, we are not suggesting that the negative effects of neonicotinoids were observed in the lab because those bees have higher levels of DWV.

In our opinion, the negative effects of neonicotinoids have been clearly established in the lab and not in the field because of two major differences between lab and field studies and the virus effect we described here.

The two main differences between lab and field studies are:

- a. in the lab, chemicals are tested at different doses, while in the field chemicals are normally tested at a certain dose which is supposed not to harm honey bees according to the results of previous lab studies (i.e., the field realistic one);
- b. in the lab, factors potentially affecting survival (e.g., DWV infection) are normally kept under control, while in the field they are normally checked only a posteriori.

For this reason, in the lab, toxicity can be assessed and an LD50 attributed to a given chemical, while in the field one can normally see only if a chemical is harmful at the tested dose or not. Therefore, both in the lab and in the field, if pesticide concentration is low one can observe satisfactory survival whereas, if pesticide concentration is high one can observe reduced survival. Our analysis shows that, if pesticide concentration is neither very low or very high, and viral infection is not controlled (as it normally happens in field studies), the survival of virus infected bees can be either short or long depending on their intrinsic conditions; at colony level, this in turn will generate results that are really difficult to predict as the published results would indicate.

We better explained this critical point in the revised version of the manuscript (see lines 222-233). Anyway, we now write that “our results may explain the contrasting results”.

Besides, Woodcock et al.’s results are inaccurately reported in lines 269-271. Unlike stated by the authors, in Hungary where negative effects were observed, Varroa mite levels were (almost) equally low as in Germany where positive effects were observed.

Thank you for pointing to our attention this circumstance which actually confirms our theory that, in the presence of an immune-suppressing virus (or the vector of it: *Varroa destructor*), honey bee survival (and, in turn, colony stability) can be either very bad or not too bad/normal depending on other uncontrollable factors. We have now corrected our wrong interpretation of Woodcock et al.’s data.

Also, the authors cite Osterman et al. (2019) as an example of a study where DWV prevalence and/or mite infestation was low that showed no adverse neonicotinoid effects. However, Osterman et al. reported that at the start of the experiment in 2013, around three-quarters of the colonies were infected with DWV Type-2. In addition, studies on the interactive effects of viruses (including DWV) and pesticides on bees have come to equally contrasting results (Harwood and Dolezal, 2020) as studies assessing only effects of neonicotinoids. In fact, studies finding synergistic effects are a minority. Besides, the study by di Prisco et al. that the authors of this article cite used primers that were criticized for producing a large number of PCR artefacts dominating the quantitative signal (Osterman et al., 2019). I do not want to say that DWV (or other pathogens) and neonicotinoids (or other pesticides) cannot interact but that the evidence that DWV drives differences in observed neonicotinoid effects in different (field) studies is slim.

We are afraid that there was a misunderstanding about our message which certainly depends on the unsatisfactory explanation of results we apparently provided. We are not suggesting that the negative effects of neonicotinoids can only be observed in presence of DWV infection. In fact, in this article we do not deal with the possible synergistic interactions between neonicotinoids and DWV. Instead, we demonstrate through a structural analysis corroborated by lab studies that the presence of an immune-suppressing pathogen can change the rules of the game according to the following scheme:

1. low dose of a stressor (e.g., a neonicotinoid insecticide) both with and without virus → acceptable survival of individual honey bee and colony stability;
2. high dose of a stressor (e.g., a neonicotinoid insecticide) both with and without virus → premature death of individual honey bee and colony dwindling;
3. intermediate level of stressor (e.g., a neonicotinoid insecticide) without virus → intermediate survival of individual honey bee and colony stability if buffering is sufficient;
4. intermediate level of stressor (e.g., a neonicotinoid insecticide) with immune-suppressing virus → either low or intermediate survival of individual honey bees and either colony dwindling or stability if buffering is sufficient.

Clearly, the really interesting result is n. 4 which accounts for the fact that in presence of a given amount of neonicotinoid insecticides, if an immune-suppressing virus is present, one can have either negative results or acceptable results. In our opinion, this is exactly what recent field data show.

In the revised text we better clarified this concept (see lines 333-339).

For the above-mentioned reasons, I do not recommend acceptance of the manuscript in its current form even though the authors address a relevant question which they tackle from various aspects, which I find an interesting approach. I do not feel competent to judge whether or not the manuscript can be revised so that it reaches the required standard for Nature communications as my understanding of a core part of the study is limited.

Further comments can be found below:

L40-41: The study by Neumann et al. is already 12 years old. To get a more up-to-date and holistic overview of honeybee colony losses, I recommend the authors to read Osterman et al., 2021 who systematically reviewed reported overwintering mortality and found considerable regional differences but no consistent increase over time.

Thank you for suggesting the most relevant recent literature. We now cite Osterman et al., 2021 which is indeed more appropriate than the one we previously cited.

L51: I suggest the authors provide at least one reference showing that “negative effects of neonicotinoid insecticides have been clearly established in the laboratory”.

Thank you for this suggestion. We now cite the review by Godfray et al. to support our statement.

L244: I suggest the authors cite here also a paper that reviews impacts of neonicotinoids on bees rather than only three original research articles that are (almost) 10 years old. There are plenty of reviews available and also a few meta-analyses.

We believe that the paper by Godfray et al. that we now cite can be regarded as an authoritative review on the subject.

*Table S1: For the Swedish field studies the authors forgot to mention that *Osmia bicornis* was assessed too (Rundlöf et al., 2015). Also, they summarize the findings as follows: “No impact on honey-bees, effects on bumble bees”. It would be more accurate and consistent with the reporting of the results of the other field studies to state that there were negative effects on bumblebees and red mason bees. Also, in the same field experiment some positive effects on honeybees were found (Osterman et al., 2019).*

We corrected our comment about the results by Rundlöf et al. Thank you.

References

*Harwood, G.P., Dolezal, A.G., 2020. Pesticide–Virus Interactions in Honey Bees: Challenges and Opportunities for Understanding Drivers of Bee Declines. *Viruses* 12, 566. <https://doi.org/10.3390/v12050566>*

*Klaus, F., Tschardtke, T., Bischoff, G., Grass, I., 2021. Floral resource diversification promotes solitary bee reproduction and may offset insecticide effects – evidence from a semi-field experiment. *Ecol. Lett.* 24, 668–675. <https://doi.org/10.1111/ele.13683>*

McArt, S.H., Fersch, A.A., Milano, N.J., Truitt, L.L., Böröczky, K., 2017. High pesticide risk to honey bees despite low focal crop pollen collection during pollination of a mass blooming crop. *Sci. Rep.* 7, 1–10. <https://doi.org/10.1038/srep46554>

Osterman, J., Aizen, M.A., Biesmeijer, J.C., Bosch, J., Howlett, B.G., Inouye, D.W., Jung, C., Martins, D.J., Medel, R., Pauw, A., Seymour, C.L., Paxton, R.J., 2021. Global trends in the number and diversity of managed pollinator species. *Agric. Ecosyst. Environ.* 322, 107653. <https://doi.org/10.1016/j.agee.2021.107653>

Osterman, J., Wintermantel, D., Locke, B., Jonsson, O., Semberg, E., Onorati, P., Forsgren, E., Rosenkranz, P., Rahbek-Pedersen, T., Bommarco, R., Smith, H.G.H.G., Rundlöf, M., de Miranda, J.R.J.R., 2019. Clothianidin seed-treatment has no detectable negative impact on honeybee colonies and their pathogens. *Nat. Commun.* 10, 1–13. <https://doi.org/10.1038/s41467-019-08523-4>

Park, M.G., Blitzer, E.J., Gibbs, J., Losey, J.E., Danforth, B.N., 2015. Negative effects of pesticides on wild bee communities can be buffered by landscape context. *Proc. R. Soc. B Biol. Sci.* 282. <https://doi.org/10.1098/rspb.2015.0299>

Pilling, E.D., Jepson, P.C., 1993. Synergism between EBI fungicides and a pyrethroid insecticide in the honeybee (*Apis mellifera*). *Pestic. Sci.* 39, 293–297. <https://doi.org/10.1002/ps.2780390407>

Rundlöf, M., Andersson, G.K.S., Bommarco, R., Fries, I., Hederström, V., Herbertsson, L., Jonsson, O., Klatt, B.K., Pedersen, T.R., Yourstone, J., Smith, H.G., 2015. Seed coating with a neonicotinoid insecticide negatively affects wild bees. *Nature* 521, 77–80. <https://doi.org/10.1038/nature14420>

Sgolastra, F., Medrzycki, P., Bortolotti, L., Renzi, M.T., Tosi, S., Bogo, G., Teper, D., Porrini, C., Molowny-Horas, R., Bosch, J., 2017. Synergistic mortality between a neonicotinoid insecticide and an ergosterol-biosynthesis-inhibiting fungicide in three bee species. *Pest Manag. Sci.* 73, 1236–1243. <https://doi.org/10.1002/ps.4449>

Vaudo, A.D., Tooker, J.F., Grozinger, C.M., Patch, H.M., 2015. Bee nutrition and floral resource restoration. *Curr. Opin. Insect Sci.* 10, 133–141. <https://doi.org/10.1016/j.cois.2015.05.008>

Wernecke, A., Frommberger, M., Forster, R., Pistorius, J., 2019. Lethal effects of various tank mixtures including insecticides, fungicides and fertilizers on honey bees under laboratory, semi-field and field conditions. *J. für Verbraucherschutz und Leb.* 14, 239–249. <https://doi.org/10.1007/s00003-019-01233-5>

Wintermantel, D., Pereira-Peixoto, M.-H., Warth, N., Melcher, K., Faller, M., Feurer, J., Allan, M.J., Dean, R., Tamburini, G., Knauer, A.C., Schwarz, J.M., Albrecht, M., Klein, A.-M., 2022. Flowering resources modulate the sensitivity of bumblebees to a common fungicide. *Sci. Total Environ.* 829, 154450. <https://doi.org/10.1016/j.scitotenv.2022.154450>

Reviewer #2 (Remarks to the Author):

The paper relates experimental results obtained on honey bees colonies with a qualitative analysis of mathematical models that describe the main interactions.

I am not expert in the specifics of honey bees ecology, so my review will only be addressing the mathematical aspects and qualitative results proposed.

The manuscript appears to be technically sound, to the best of my knowledge, and provides an interesting instance of qualitative analysis tools enabling interpretation and reconciliation of apparently contradicting empirical observations.

We gratefully thank Reviewer 2 for her/his positive appreciation of our work.

I have some minor suggestions for improving clarity of presentation:

Line 109-110: "by different functions which are all decreasing to 0 because they exert a negative effect on honey bee health".

This sentence is slightly misleading as it stands.

I believe a better formulation would be: "by different monotonically decreasing functions because each factor exerts a negative effect on honey bee health."

The fact that these are decreasing to zero is perhaps a suitable modelling assumption but is unrelated to the fact that the effect is negative.

We agree and we rephrased the sentence as the Reviewer suggests (see lines 128-129).

Line 123: "if the sign of the first state variable is changed."

While I understood what authors meant I believe the sentence is unclear as it seems to imply an arbitrary manipulation of the model that may alter its significance or dynamical behaviour.

Perhaps the following formulation improves readability:

"If the model is reformulated by using as a first state variable the opposite of x_{BH} (viz. an indicator of bee unhealthiness)"

We reformulated the sentence as suggested (see lines 143-144).

Line 364: It seems to me that matrix M is Hurwitz by assumption (as is similar to J), not because of its sign pattern.

We removed the misleading comment about the sign pattern (see line 437 in the revised manuscript).

Overall, while I found the phenomena modelled and described in the paper to be a convincing paradigm of a situation in which qualitative analysis is important to interpret and validate field studies.

We thank the Reviewer for capturing one of the messages of our manuscript in such a clear way.

Reviewers' Comments:

Reviewer #1:

Remarks to the Author:

I thank the authors for their extensive answers to my queries. They satisfactorily responded to most points (as far as I can tell without a background in mathematical modeling). I have however some remarks:

- The authors claim that adding more interactive effects to their mathematical model would not change their conclusions. It is also my understanding that the principal conclusions on how a pesticide can affect bees in presence/absence of a virus would not change if for instance potential interactive effects between different pesticides were considered. However, similar conclusions may be reached for pesticide-pesticide interactions (in the sense that one stressor/pesticide impacts the impact of another stressor/pesticide) if they examined it in the same way as pesticide-pathogen interactions. Similarly, if pesticide exposure or factors affecting pesticide exposure were more explicitly included in the model, potentially it would show how these drive differences between field studies. If the goal of the study was to show that pathogens can affect the observed effects of pesticides, it was achieved, but the study goal was formulated broader.
- The authors continue to claim that Fig. 3A shows a unimodal longevity distribution for early-season honeybees and a bimodal longevity distribution for late-season honeybees. However, I still don't see it. I agree with what they wrote in response to me but I don't see a difference in peaks/modes. In fact, if we ignore age=4 days then both early-season and late-season honeybees show a unimodal distribution. If we do not ignore age=4 days, then we could with some good will argue for a bimodal distribution, in both cases (and the argument would not be stronger for late-season honeybees). The difference does not lie in the number of modes/peaks but in the variance, which is larger for late-season than early-season honeybees. One may argue that this also supports their conclusions from the mathematical model that outcomes can be more different if a virus is present (even though it is quite indirect which it is irrespectively of if the argument is based on the number of modes or the variance).
- I do understand that density plots require an additional assumption. However, to ensure that all bars are visible, one can simply overlay histograms see e.g Wintermantel, D., Locke, B., Andersson, G.K.S., Semberg, E., Forsgren, E., Osterman, J., Rahbek Pedersen, T., Bommarco, R., Smith, H.G., Rundlöf, M., de Miranda, J.R., 2018. Field-level clothianidin exposure affects bumblebees but generally not their pathogens. *Nat. Commun.* 9. <https://doi.org/10.1038/s41467-018-07914-3>
- Fig. S1A does not show whether the trends that are observed in Fig. 2 are observable if better discrimination between different birth dates is done. It only shows that DWV increases as the season progresses but it does not show how this affects longevity.

Reviewer #2:

Remarks to the Author:

I am satisfied with the way my comments were addressed.

Below we report, in italic, our point by point response to the reviewers' comments.

Reviewer #1 (Remarks to the Author):

I thank the authors for their extensive answers to my queries. They satisfactorily responded to most points (as far as I can tell without a background in mathematical modeling). I have however some remarks:

- The authors claim that adding more interactive effects to their mathematical model would not change their conclusions. It is also my understanding that the principal conclusions on how a pesticide can affect bees in presence/absence of a virus would not change if for instance potential interactive effects between different pesticides were considered. However, similar conclusions may be reached for pesticide-pesticide interactions (in the sense that one stressor/pesticide impacts the impact of another stressor/pesticide) if they examined it in the same way as pesticide-pathogen interactions. Similarly, if pesticide exposure or factors affecting pesticide exposure were more explicitly included in the model, potentially it would show how these drive differences between field studies. If the goal of the study was to show that pathogens can affect the observed effects of pesticides, it was achieved, but the study goal was formulated broader.

We confirm that our structural analysis predicts that adding more interactive effects between pesticides would not change our conclusions about the effect of an immune-suppressive virus on the system's response to toxic chemicals.

Instead, we do not think that similar conclusions would be reached if we examined pesticide-pesticide interactions in the same way as pesticide-pathogen interactions.

The reason why we can confidently state so without the need of further modelling, is that, among the considered stressors, only the pathogen has got the capacity of disrupting the system that is contrasting the pathogen itself, thus introducing a dangerous positive feedback loop in the system. In our conceptual model, this peculiarity of the pathogen is captured by the closed loop formed by arrows "m" and "j" between "immunity" and "deformed wing virus" in Fig. 1A. In mathematical terms, this can be clearly seen from the equations: functions h , which convey the effect of the virus, are increasing with respect to x_{VI} (the state variable associated with the virus). When the parameter ε , associated with the immune-suppressing potential of the virus, is large enough, the presence of function h_{VI} in the equation describing the time evolution of x_{VI} yields the ability of the virus to increase its own effect.

We have now added a sentence to clarify this fundamental difference between the virus and pesticides (see lines 212-218).

As far as we know, there is no other stressor with this capacity to generate positive feedback. In particular, to date, no chemical has been proved to be capable of impairing the system controlling the response to chemicals (e.g. a pesticide impairing the detoxification system). Therefore, other pesticide interactions will only alter the magnitude of the pesticide effect which will not alter the conclusions.

Although some studies highlighted the effect of certain pesticides on the expression of a few detoxification genes in the honey bee, this effect, normally consisting of an up-regulation of some genes after exposure to pesticides, does not suggest that such pesticides can impair detoxification, but should rather be regarded as evidence of a well-functioning homeostatic system that reacts to

intoxication by triggering a physiological mechanism aimed at reducing the concentration of the toxic chemical.

In any case, based on our analysis we can hypothesize that the possible anti-detoxification activity of a pesticide (perhaps still to be discovered) may cause an effect similar to that reported here for the pathogenic virus DWV. At present this possibility is purely speculative but, due to the important implications of such a hypothetical effect for honey bee survival, we added a comment about this point in the revised version of our manuscript (see lines 347-360).

Finally, we agree with the reviewer that pesticide exposure could drive possible differences between field studies and by no means we want to suggest that all the possible differences between field studies are related to the effect we describe in our manuscript, thus excluding other possible explanations such as pesticide exposure.

We simply state that:

a. the presence of an immune-suppressing virus can cause bistability

b. bistability is related to unpredictable effects

c. such unpredictable effects can explain the observed differences between field studies.

We now added a comment to clarify that c) does not imply that all the possible differences between field studies are due to the bistability caused by the immune-suppressing virus (see lines 380-385).

We also changed the title to clarify that our conceptual framework facilitates the interpretation of contradictory field results, but we do not claim that we can explain everything through it.

In any case, it is worth noting that the most puzzling results regarding the effect of neonicotinoids under field conditions (i.e. those obtained by Woodcock et al., who reported a country specific effect of neonicotinoids) were obtained with a robust experimental plan. For this reason, although variable pesticide exposure cannot be excluded, in our opinion this is not the most plausible explanation for the observed discrepancy between the results obtained in different countries.

- The authors continue to claim that Fig. 3A shows a unimodal longevity distribution for early-season honeybees and a bimodal longevity distribution for late-season honeybees. However, I still don't see it. I agree with what they wrote in response to me but I don't see a difference in peaks/modes. In fact, if we ignore age=4 days then both early-season and late-season honeybees show a unimodal distribution. If we do not ignore age=4 days, then we could with some good will argue for a bimodal distribution, in both cases (and the argument would not be stronger for late-season honeybees). The difference does not lie in the number of modes/peaks but in the variance, which is larger for late-season than early-season honeybees. One may argue that this also supports their conclusions from the mathematical model that outcomes can be more different if a virus is present (even though it is quite indirect which it is irrespectively of if the argument is based on the number of modes or the variance).

We understand the reviewer's perplexity and thank her/him for pointing to our attention a critical aspect that needs to be clarified. Below we try to clarify that point and illustrate the revisions we made to the manuscript so that the readers can also follow this.

The reviewer is concerned by the lack of a clearer difference in the number of modes of the longevity distributions between treated and control samples. Indeed, a small number of bees dying

very early is present also in the “virtually virus free” experimental groups (highlighted with orange-red colors in the various panels of fig. 3).

On the other hand, the reviewer recognizes that both the graphs and the table clearly show that the dispersion of data is greatly affected by the presence of the virus, with the IQR nearly doubling its value in the presence of the virus, rather consistently across treatments (see table 1).

In fact, it is the data distribution (rather than the number of modes) that matters the most, for the following reasons.

We used our model to predict honey bee health as affected by a number of interacting stress factors and showed that, in the presence of a sufficiently high level of an immune-suppressing virus, any stressor that is present at an intermediate level (not too high or too low) can cause honey bee health to set at any of two different equilibria: either high or low. This would imply that, under these conditions, in the presence of a large enough viral load, a bimodal distribution of honey bee health should be seen and the variability of data should increase steeply.

To confirm this conclusion, we wanted to use real-world data. To this aim we exploited a rich data set, including the results of many lab experiments carried out in a standardized manner over a long period of time in a single lab. Despite the excellent quality of data, we had to cope with a number of practical problems.

- For example, to estimate honey bee health we had to choose a proxy; we used longevity which, however, is not the same as health, because it is influenced by other factors.
- Furthermore, since virus free bees are rather difficult to obtain nowadays on this planet, we had to resort to bees collected when viral circulation is low, and, even under those conditions, infection level is never nil so that we found infected specimens in the control groups (see our previous response).
- Also, apart from the experiment whereby we directly infected bees, we could not control the level of viral infection (e.g. the number of DWV genome copies per bee) but only the virus infection state (e.g. virus infected or virus free), and, as reported in the literature and prescribed by our model, immune-suppression and bistability ensue only if the virus is above a certain level.

As a result of these and other problems, bimodality was not always clearly seen.

On the other hand, the effect on data variability was always clearly visible and perfectly in line with our hypothesis.

Apparently, the initial wording of our manuscript attracted the attention more on bimodality than on dispersion, and this caused the perplexity of the reviewer. To comply with the request, we decided not to refer to ‘bimodality’ but to ‘dispersion’; we believe that the new version of the manuscript is now clearer under this respect (see lines 248-255).

- I do understand that density plots require an additional assumption. However, to ensure that all bars are visible, one can simply overlay histograms see e.g Wintermantel, D., Locke, B., Andersson, G.K.S., Semberg, E., Forsgren, E., Osterman, J., Rahbek Pedersen, T., Bommarco, R., Smith, H.G., Rundlöf, M., de Miranda, J.R., 2018. Field-level clothianidin exposure affects bumblebees but generally not their pathogens. Nat. Commun. 9. <https://doi.org/10.1038/s41467-018-07914-3>

We now use 2D histograms and all bars are visible. We are now convinced that the effect we wanted to underline with this graph is clearly visible and we thank the reviewer for the suggestion.

- Fig. S1A does not show whether the trends that are observed in Fig. 2 are observable if better discrimination between different birth dates is done. It only shows that DWV increases as the season progresses but it does not show how this affects longevity.

Fig.S1A only shows that the decision to consider the bees emerging before mid-July as virtually virus free and those emerging later as virus infected is not only based on convincing literature data, but also on the new data reported in this article.

To underline this concept, we decided to add another reference to an article demonstrating that, under the conditions of the area where the experiments were carried out, virus prevalence and infection level are low early in the season and high late in the season (see lines 248-255).

It is worth noting, however, that the fact that DWV affects longevity regardless of the season is clearly proved by the second experiment whereby bees emerging in the same period of time were exposed or not to the virus. This shows that It is not the season per se, but the viral infection that is linked to the season, that determines bistability.

Reviewer #2 (Remarks to the Author):

I am satisfied with the way my comments were addressed.

We thank the reviewer for the useful comments previously received and for the positive final assessment.